# Autophagy promotes jasmonate-mediated defense against nematodes

Jinping Zou[1,8], Xinlin Chen[1,8], Chenxu Liu[1], Mingyue Guo[1], Mukesh Kumar Kanwar[1], Zhenyu Qi[2,3,4], Ping Yang[3], Guanghui Wang[5], Yan Bao[6], Diane C. Bassham [7], Jingquan Yu [1,2,4] & Jie Zhou [1,2,4,5] ✉

Autophagy, as an intracellular degradation system, plays a critical role in plant immunity. However, the involvement of autophagy in the plant immune system and its function in plant nematode resistance are largely unknown. Here, we show that root-knot nematode (RKN; *Meloidogyne incognita*) infection induces autophagy in tomato (*Solanum lycopersicum*) and different *atg* mutants exhibit high sensitivity to RKNs. The jasmonate (JA) signaling negative regulators JASMONATE-ASSOCIATED MYC2-LIKE 1 (JAM1), JAM2 and JAM3 interact with ATG8s via an ATG8-interacting motif (AIM), and JAM1 is degraded by autophagy during RKN infection. JAM1 impairs the formation of a transcriptional activation complex between ETHYLENE RESPONSE FACTOR 1 (ERF1) and MEDIATOR 25 (MED25) and interferes with transcriptional regulation of JA-mediated defense-related genes by ERF1. Furthermore, ERF1 acts in a positive feedback loop and regulates autophagy activity by transcriptionally activating *ATG* expression in response to RKN infection. Therefore, autophagy promotes JA-mediated defense against RKNs via forming a positive feedback circuit in the degradation of JAMs and transcriptional activation by ERF1.

Autophagy is a highly conserved self-degrading process that breaks down unnecessary damaged components in eukaryotes and recycles cellular nutrients. In plant cells, autophagy maintains homeostasis under normal conditions and is a survival mechanism under external stress[1]. Autophagy also plays a crucial role in plant nutrient deficiencies and abiotic stress responses, including heat, salt, drought and darkness[2-4]. In 2005, Liu et al. were the first to discover that autophagy positively regulates plant immunity during N protein-mediated defense against tobacco mosaic virus (TMV)[5]. Subsequently, autophagy was increasingly demonstrated to be involved in plant defense and disease resistance responses to viral, bacterial and fungal pathogens[6-8]. In general, it is widely believed that autophagy plays dual

and pleiotropic roles in plant immune response[9,10]. On the one hand, a large number of studies have shown that autophagy eliminates and inhibits pathogen and virus infection by direct identification and degradation of microbe components so as to achieve the purpose of disease resistance[11,12]. Geminivirus Cotton leaf curl Multan virus (CLCuMuV) infection activates autophagy, and that autophagy targets the virulence factor βC1 for degradation through its interaction with the key autophagy-related protein 8 (ATG8) and improves host immunity[11]. Selective autophagy limits cauliflower mosaic virus (CaMV) infection through the removal of viral capsid protein and particles by autophagy cargo receptor NEIGHBOR OF BRCA1 (NBR1)[12]. On the other hand, some phytopathogens directly interfere with ATGs and

[1]Department of Horticulture, Zhejiang Provincial Key Laboratory of Horticultural Plant Integrative Biology, Zhejiang University, Yuhangtang Road 866, 310058 Hangzhou, China. [2]Hainan Institute, Zhejiang University, 572000 Sanya, China. [3]Agricultural Experiment Station, Zhejiang University, 310058 Hangzhou, China. [4]Key Laboratory of Horticultural Plants Growth, Development and Quality Improvement, Ministry of Agriculture and Rural Affairs of China, Yuhangtang Road 866, 310058 Hangzhou, China. [5]Shandong (Linyi) Institute of Modern Agriculture, Zhejiang University, 276000 Linyi, China. [6]Shanghai Collaborative Innovation Center of Agri-Seeds, Joint Center for Single Cell Biology, School of Agriculture and Biology, Shanghai Jiao Tong University, 200240 Shanghai, China. [7]Department of Genetics, Development and Cell Biology, Iowa State University, Ames, IA 50011, USA. [8]These authors contributed equally: Jinping Zou, Xinlin Chen. ✉e-mail: jie@zju.edu.cn

manipulate autophagy to promote their proliferation and virulence in plant cells[13–16]. *Pseudomonas syringae* pv. *tomato* (*Pto*) DC3000 utilizes type-III effector HopM1 to stimulate autophagy for proteasome degradation and the benefit of infection in *Arabidopsis*[14]. The other *Pto* DC3000 secretory effectors, HrpZ1 and HopF3, enhance or suppress autophagy to promote infection by interacting with ATG4 or ATG8, while AvrPtoB affects ATG1 kinase activity to enhance bacterial virulence[15]. *Xanthomonas campestris* pv. *vesicatoria* (*Xcv*) suppresses host autophagy and promotes infection by utilizing type-III effector XopL, which interacts with and degrades host autophagy component SH3P2 via its E3 ligase activity to promote infection[16]. Intriguingly, NBR1-mediated selective autophagy plays antibacterial roles by degradation of the effectors or the turnover of ubiquitinated substrates during infection, suggesting a complex antagonistic interplay between effectors and host autophagy machinery[15,16].

A set of ATG proteins have been identified to be involved in various stages of autophagy[17]. Among these ATGs, the ubiquitin-like protein ATG8 plays a central role in plant autophagy. The ATG8 protein is conjugated to the membrane lipid phosphatidylethanolamine (PE) in a ubiquitin-like conjugation reaction that is important for autophagosome formation[18]. ATG8 also plays an important role in selective autophagy by interacting with various autophagy adapters and receptors to recruit specific cargos for degradation. The ATG8-interacting proteins usually contain ATG8-interacting motifs (AIMs, W/F/YXXL/I/V) or ubiquitin-interacting motifs (UIMs) for ATG8 binding[19]. Various ATG8-binding proteins act as selective autophagy receptors, mediating the delivery of specific target cargos to autophagosomes for degradation[20–22]. Several studies have shown that NBR1 binds ubiquitin through a C-terminal UBA domain and interacts with various ATG8 proteins via an evolutionarily conserved AIM to play a key role in the plant response to abiotic stresses such as heat, drought, salt or oxidative stress, mediating the degradation of ubiquitinated cargos or protein aggregates[20,21]. In addition, NBR1 has also been shown to provide antiviral effects in cauliflower mosaic virus (CaMV) infection, mediating their autophagic degradation in the plant defense response[12]. Several other studies demonstrate that autophagy can control hormone levels and signaling in basal resistance to pathogens and viruses and to regulate defense- and disease-related cell death[23–25].

Jasmonic acid (JA) is a major defense phytohormone that plays a pivotal role in regulating plant defense responses to mechanical wounding, insect attack and pathogen infection[26–28]. Upon mechanical wounding or insect/pathogen attack, JA biosynthesis turned on rapidly. The bioactive jasmonoyl-L-isoleucine (JA-Ile) is perceived by the CORONATINE INSENSITIVE 1 (COI1)-jasmonate ZIM-domain (JAZ) complex, leading to the degradation of JAZ repressor proteins via the 26S proteasome and release of downstream transcription factors to turn on various JA-responsive genes[29]. The JA signaling pathway consists of two branches; the basic helix-loop-helix (bHLH) protein (MYC) branch coupled to wounding and defense against insect herbivores and the ethylene-responsive factor (ERF) branch that is associated with enhanced resistance to necrotrophic pathogens[30]. As a core transcription factor in the JA signaling pathway, MYC2 interacts either with the transcriptional suppressor JAZ to perform its transcriptional inhibitory function or with the transcriptional activator MEDIATOR 25 (MED25) to perform its transcriptional activating function[31,32]. In addition to MYC2, ERFs are also key factors in the JA signaling pathway and are involved in the transcriptional regulation of various biological processes in plant stress responses[33–35]. JA and ethylene are usually produced simultaneously during pathogen infection and synergistically regulate resistance defense signaling pathways[36–38]. In addition, autophagy plays a key role in plant resistance to necrotrophic fungal pathogens by inhibiting pathogen-induced cell death and disturbing the hormonal balance via the antagonism between JA and SA signaling[25]; in turn, JA-related WRKY genes have been implicated as mediating autophagy gene expression during fungal pathogen

infection[23]. However, there are few reports on the crosstalk between autophagy and the JA signaling pathway, and the mechanism of their signaling regulation is still unclear.

Root-knot nematodes (RKNs, *Meloidogyne* spp.) are plant-parasitic nematodes, such as *M. arenaria*, *M. javanica*, *M. incognita* and *M. hapla*, with a wide host range and which cause huge economic losses to crops[39–41]. In response to nematode invasion, plants have evolved various defense strategies to induce immune responses[42]. Notably, recent studies have found that the JA-dependent signaling pathway plays a critical role in pathogen-associated molecular pattern (PAMP)-triggered immunity (PTI) and effector-triggered immunity (ETI) against nematodes and necrotrophic pathogens[43–47]. In tomato, previous studies have reported that JA-dependent signaling is not involved in *Mi-1*-mediated defense, while an intact JA signaling pathway is required for tomato resistance to RKNs[43,44]. In rice (*Oryza sativa*), exogenous ethephon (ET) and methyl JA (MeJA) upregulated the expression of *OsPR1a* and *OsPR1b* genes at the initial stage of *M. graminicola* infection, thereby positively regulating the systemic defense of rice against nematode parasitism[45]. Furthermore, JA-responsive genes, such as *PLANT DEFENSIN 1.2* (*PDF1.2*) and *PROTEINASE INHIBITORS* (*PIs*), are involved in JA-induced resistance against RKNs[48]. Although the JA signaling pathway occupies a crucial position in plant RKN resistance, its regulatory mechanism is largely unknown.

In this study, we show that tomato (*Solanum lycopersicum*) *atg* mutants exhibit high sensitivity to RKNs (*M. incognita*). The negative regulators of JA signaling, JASMONATE-ASSOCIATED MYC2-LIKE 1 (JAM1), JAM2, and JAM3, interact with ATG8s through the AIM domain and are degraded by autophagy in response to RKN infection. In addition, JAMs impair the formation of ERF1 and MED25 transcriptional activation complexes and interfere with ERF1 transcriptional regulation of JA-mediated defense-related genes. Conversely, ERF1 positively feedback-regulates autophagy by transcriptionally regulating *ATG* expression. Our results revealed that autophagy promotes JA-mediated defense against RKNs by forming a positive feedback loop to degrade JAMs and activate ERF1 transcriptional activity, providing a paradigm for the functional study of plant autophagy and JA signaling pathway in the regulation of pathogen infection processes.

## Results

### Autophagy is essential for tomato nematode resistance

To elucidate whether autophagy is involved in plant defense against RKNs, we analyzed the expression patterns of 23 *ATGs* in wild-type tomato plants (Ailsa Craig, AC) in response to RKNs at different time points. As shown in Fig. 1a, RKN infection induced differential expression of *ATGs*, most of which were upregulated after infection, especially at 36 h post-infection (hpi). To confirm autophagosome production, we used green fluorescent protein (GFP)-tagged ATG8f (GFP-ATG8f) as a marker to examine the formation of autophagosomes after RKN infection. As expected, the GFP-ATG8f labeling results showed an increased number of autophagosomes in AC roots after RKN infection and reached a maximum at 36 hpi. We also monitored autophagosomal structures during Concanamycin A (ConA) treatment, which blocks the vacuolar-type ATPases on the tonoplast membrane responsible for vacuolar acidification[49]. ConA treatment blocked autophagic body degradation, which not only enhanced the microscopic detection of autophagosomes but also stabilized their content (Fig. 1b, c). The formation of ATG8-phosphatidylethanolamine (PE) conjugates can be used as a marker of autophagy activation using western blotting[3]. A weak ATG8-PE band was detected in WT roots under normal conditions and the signal increased after RKN infection (Fig. 1d). We also detected ATG8-PE bands in WT and *atg7* mutants with or without RKN infection after ConA treatment. As shown in Fig. 1e, RKN-induced ATG8-PE bands were compromised in *atg7* roots with or without ConA treatment, however, RKN-induced ATG8 and ATG8-PE bands were both further accumulated in WT roots after ConA

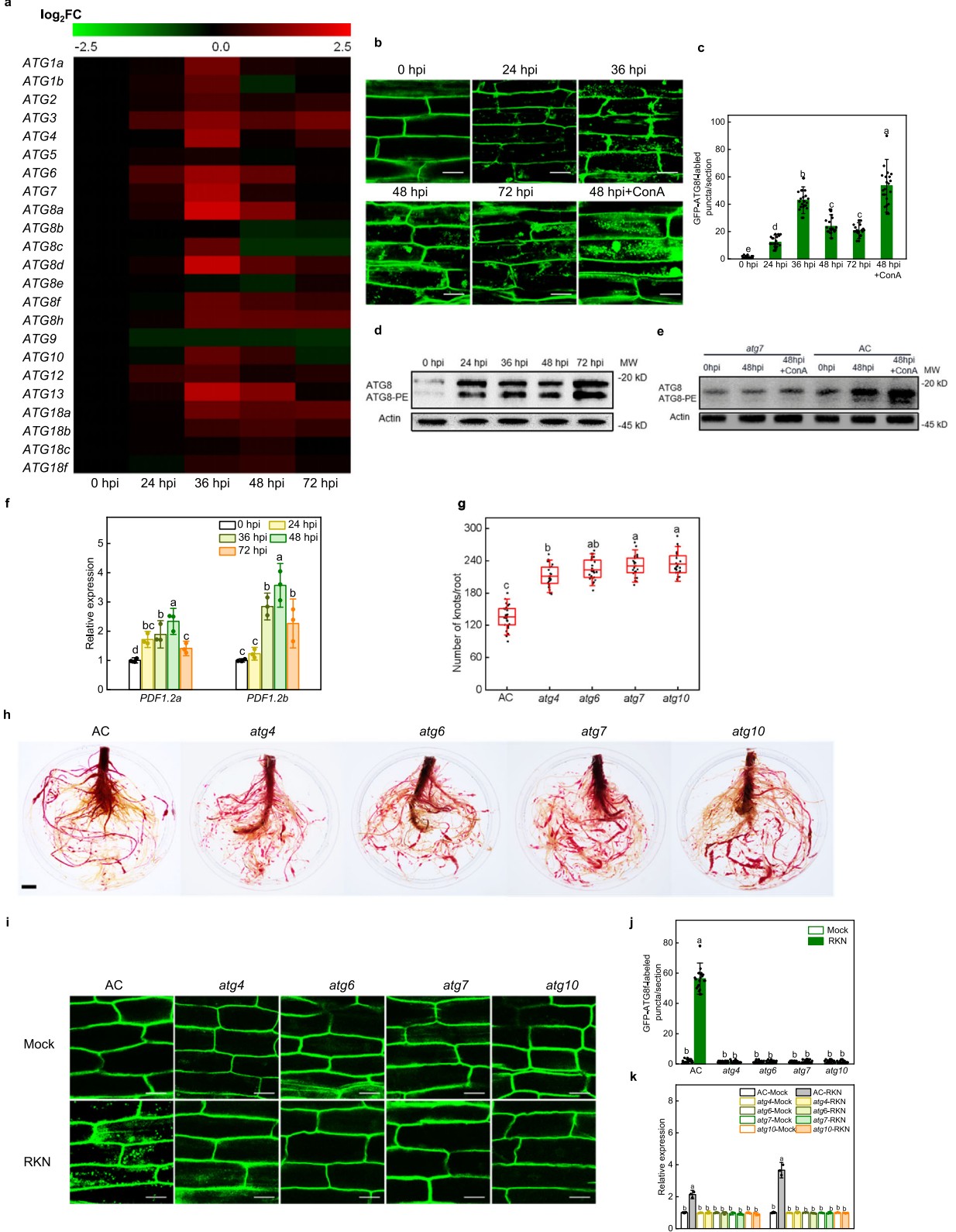

treatment. These results indicate that the infection of RKNs induces autophagy in tomato roots.

Next, we analyzed the expression levels of JA marker and responsive genes, *PDF1.2a* and *PDF1.2b*, in tomato after RKN infection. As shown in Fig. 1f, the expression levels of both *PDF1.2a* and *PDF1.2b* were induced from 36 to 72 hpi with RKNs and reached the highest expression level at 48 hpi.

To comprehensively understand the function of autophagy in tomato RKN defense, we detected root knots in four different *atg* mutants. Wild-type AC and four *atg* mutants were infected with RKNs and the phenotypes were observed after 5 weeks. Root knot numbers in these *atg* mutants were significantly higher than wild-type AC seedlings, showing higher susceptibility to RKN infection. There were approximately 135.4 knots per plant in AC roots, and the knots were

**Fig. 1 | Autophagy plays an active role in plant defense against *Meloidogyne incognita*. a** Heatmap of *ATGs* expression at different time points after RKN infection of Ailsa Craig (AC) tomato roots. The labels 0 h post-infection (hpi), 24 hpi, 36 hpi, 48 hpi, and 72 hpi at the bottom indicate the time point of RKN infection. Transcript levels were determined using RT-qPCR, and cluster analysis was performed by MeV version 4.9, and data were transformed by $\log_2$-fold change (FC). The color bar at the top indicates expression levels. **b** The formation of autophagosomes in the roots. The direct fluorescence of GFP-ATG8f was detected in the roots by confocal fluorescence microscopic analysis with or without RKN infection. The seedlings were pretreated with 1 μM the vacuolar-type ATPase inhibitor Concanamycin A (ConA) for 3 h prior to confocal fluorescence microscopy. Bars, 25 μm. **c** Quantification of (**b**). The number of autophagosomes per image was quantified to calculate the autophagic activity relative to wild-type control plants, which was set to '1'. Error bars represent SD; data represent the mean ± SD ($n = 20$ samples, individual dots). The experiments were repeated twice with similar results. **d**, **e** Immunoblot detection of ATG8-PE. Changes in ATG8-PE after infection with RKNs in WT plants (**d**). Induction of autophagy by RKN infection and ConA treatment (**e**). ATG8 and ATG8-PE are the nonlipidated and lipidated forms of ATG8, respectively. The Actin protein was used as a loading control for the western blotting analysis. The experiments were repeated twice with similar results (**d**, **e**). **f** The expression of *PDF1.2a* and *PDF1.2b* after RKN infection in AC plants. Error bars represent SD; data represent the mean ± SD ($n = 3$ biological replicates, individual dots). **g** The number of root knots of plants at 5 weeks after infection with RKNs. Data are presented as boxplots, with each dot representing the datapoint of one biological replicate ($n = 25$ plants). For the boxplots, the central line indicates the mean value, the bounds of the box show the 1st and 3rd quartile, and the whiskers indicate 1.5× interquartile range between the 1st and 3rd quartile. **h** Phenotype of RKN reproduction in AC and *atg* mutants using acid fuchsin staining after RKN infection. Bar, 1 cm. The experiments were repeated twice with similar results. **i, j** The formation of autophagosomes after RKN infection at 36 h in the roots of AC and *atg* mutants. The direct fluorescence of GFP-ATG8f (**i**) was detected in the roots and the number of GFP-ATG8f-labeled puncta (**j**) per image in (**i**). Bars, 25 μm. The number of autophagosomes in each image was quantified, and the autophagic activity of wild-type control plants was set as '1'. Error bars represent SD; data represent the mean ± SD ($n = 20$ samples, individual dots). The experiments were repeated twice with similar results. **k** Expression of *PDF1.2a* and *PDF1.2b* after RKN infection at 48 h in AC and *atg* mutants. Error bars represent SD; data represent the mean ± SD ($n = 3$ biological replicates, individual dots). Different letters above bars indicate a significant difference at the $P < 0.05$ level by one-way ANOVA with Tukey's multiple comparisons test. Exact $P$-values of statistic tests are provided in the Source data file.

increased by 56.0%, 64.9%, 70.3% and 72.8% in the roots of *atg4*, *atg6*, *atg7* and *atg10* plants compared to AC roots, respectively (Fig. 1g, h). The formation of GFP-ATG8f-labeled punctate autophagosomes was also compromised in *atg* mutants at 36 hpi with RKNs (Fig. 1i, j). Interestingly, RKN defense genes *PDF1.2a* and *PDF1.2b* were both induced in wild-type AC but were not significantly changed in *atg* mutants after RKN infection (Fig. 1k). These results provide evidence that autophagy plays a positive role in regulating RKN resistance in tomato.

### JAMs interact with ATG8a by the ATG8-interaction motif

ATG8 is a core protein of autophagy that selects cargoes for degradation[50,51]. In tomato, seven ATG8 homologs have been identified, including ATG8a-f and ATG8h. According to amino acid sequence alignment, the ATG8 proteins in tomato can be divided into three subgroups[51,52]. The first group consists of ATG8a, ATG8c, ATG8d and ATG8f, the second group consists of ATG8b and ATG8e, and the third group consists of ATG8h in tomato. To investigate the potential targets of ATG8, we used ATG8a as a bait to screen for interacting proteins by yeast two-hybrid (Y2H) library screening. In total, we identified more than 50 ATG8a-interacting proteins. Among these candidates, we found a bHLH type transcription factor named jasmonate-associated MYC2-like 1 (JAM1, Solyc01g096050) as a potential gene of interest. By homologous sequence alignment from the Tomato Genome Sequencing Project, JAM1 has two homologous proteins, JAM2 (Solyc05g050560) and JAM3 (Solyc06g083980), in tomato. A phylogenetic tree built from the alignment of these three proteins with the previously identified Arabidopsis JAMs revealed the evolutionary distances between the sequences (Supplementary Fig. 1). By the analysis of amino acid sequences, we found that the JAMs contained two AIMs at their N-terminus (AIM1) and C-terminus (AIM2) (Fig. 2a). Therefore, we first tested the interaction between ATG8a and JAMs by using Y2H. As shown in Fig. 2b, ATG8a interacted with JAM1, JAM2 and JAM3. Interestingly, upon mutation of AIM1 in JAM1 and JAM2 (from FWQI to AWQA), and JAM3 (from YWQV to AWQA), the JAM-ATG8a interactions were all disrupted, whereas, upon mutation of AIM2 in JAM1, JAM2 and JAM3 (from FYAL to AYAA), the JAMs still associated with ATG8a, implying that the N-terminal AIM domain was critical for the JAM-ATG8a interactions (Fig. 2b). Moreover, we tested the interaction between ATG8a homologous proteins (ATG8b-f and ATG8h) and JAMs or JAM AIM mutants, showing that the first group ATG8d and ATG8f, and the third group ATG8h also interact with JAM1, JAM2 and JAM3 via the AIM1 motif (Supplementary Fig. 2).

These results suggest that the interaction between JAMs and ATG8s is not only related to the N-terminal AIM domain of JAMs but also is related to ATG8 structures.

JAMs are transcription factors, however, these proteins are localized in both the nucleus and cytoplasm (Supplementary Fig. 3). A bimolecular fluorescence complementation (BiFC) assay in which JAM1-nYFP and cYFP-ATG8a were co-expressed in tobacco leaves, further demonstrated JAM1-ATG8a interaction in vivo (Fig. 2c). mCherry-ATG8f fusion protein was used to detect the formation of ATG8-labeled punctate fluorescent signals in plant cells likely representing autophagosomes (Fig. 2c). The detected BiFC signals from the interaction of JAM1-nYFP and cYFP-ATG8a and mCherry-ATG8f signals were both largely dispersed with few punctate fluorescent structures under normal conditions. After treatment with AZD8055 (15 μM) to induce autophagy[17], the numbers of punctate fluorescent signals of JAM1-ATG8a BiFC and mCherry-ATG8f were both sharply increased and were strongly overlapped (Fig. 2c). As expected, when AIM1 mutated JAM1 (JAM1$^{mAIM1}$)-nYFP was co-expressed with cYFP-ATG8a, no BiFC signals or punctate structures were observed in the infiltrated tobacco leaves. Likewise, similar results were also discovered in co-expressing JAM2-nYFP and JAM3-nYFP with cYFP-ATG8a in tobacco leaves (Supplementary Fig. 4). The interaction of ATG8a and JAM1 was further confirmed by co-immunoprecipitation (Co-IP). MYC-ATG8a proteins were detected in the immunoprecipitated JAM1 protein complex, but MYC-ATG8a protein failed to be co-immunoprecipitated by AIM1 mutated JAM1 (Fig. 2d). These results suggest that the N-terminal AIM domain determines the binding affinity of JAM1, JAM2 and JAM3 to ATG8s.

### JAMs are specifically degraded by autophagy and negatively regulate RKN resistance

To determine whether JAM1, JAM2 and JAM3 are involved in tomato RKN defense, we first analyzed the expression levels of *JAM1*, *JAM2* and *JAM3* at different time points after RKN infection. Interestingly, the expression of *JAMs* was all induced after RKN infection (Fig. 3a). In contrast, the accumulation of JAM1 protein was gradually decreased in wild-type AC roots after RKN infection (Fig. 3b). We then tested whether autophagy is involved in JAM degradation in tomato response to RKN infection, for which we compared JAM1 protein levels in roots of AC plants and *atg7* mutants using the ubiquitin proteasome inhibitor MG132, the vacuolar-type ATPase inhibitor ConA and the autophagy inducer AZD8055 with or without RKN infection. As shown in Fig. 3c, the accumulation of JAM1 protein in AC roots was significantly

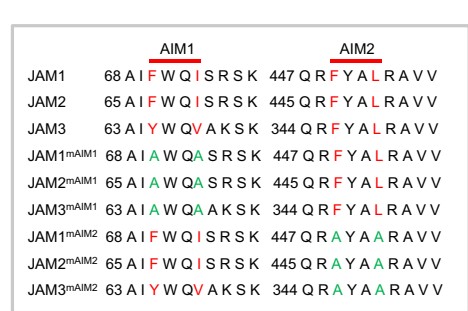

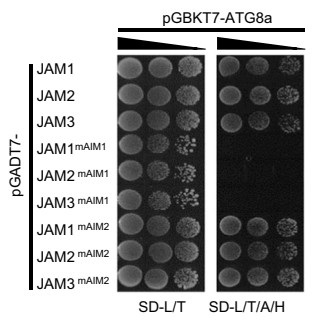

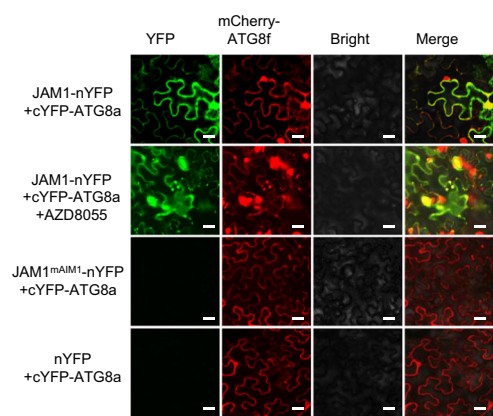

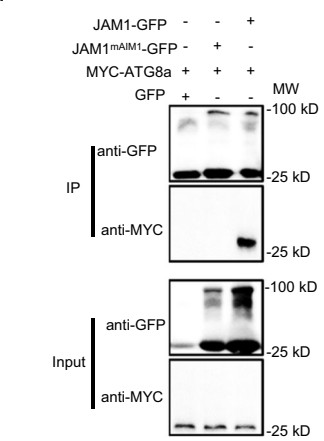

**Fig. 2 | JAM proteins interact with ATG8a via AIM. a** Two conserved AIMs (AIM1 and AIM2) and mutant AIMs (mAIM1 and mAIM2) of tomato JAM1, JAM2 and JAM3. Mutation sites are highlighted in green. **b** Yeast two-hybrid assays of the interaction between ATG8a and JAM1/2/3, JAM1/2/3 ^mAIM1 and JAM1/2/3 ^mAIM2. Protein-protein interactions were evaluated by the different concentrations of yeast cells growth on selective media lacking Leu (L), Trp (T), Ade (A), and His (H) (SD-L/T/A/H). **c** Bimolecular fluorescence complementation assays of the interaction between ATG8a and JAM1 or JAM1^mAIM1. mCherry-ATG8f acts as a marker for autophagosomes. The YFP and mCherry signals were observed under confocal microscopy 48 h after infiltration; then the tobacco was treated with the autophagy activator AZD8055 (15 μM) for 3 h to detect the accumulation of autophagosomes. Bars, 50 μm. **d** Co-immunoprecipitation assays of the interaction between ATG8a and JAM1 or JAM1^mAIM1. Total proteins were extracted from *N. benthamiana* leaves transiently expressing MYC-ATG8a fusions, JAM1-GFP, JAM1^mAIM1-GFP, or empty GFP after 48 h of infiltration. The extracted proteins were immunoprecipitated with an anti-GFP antibody and the presence of MYC-ATG8a in the immune complex was determined by immunoblot with anti-MYC antibody. The experiments were repeated twice with similar results (**b–d**).

increased after ConA treatment but was significantly reduced after AZD8055 treatment, while MG132 treatment did not change the JAM1 protein level in AC roots. The JAM1 content in *atg7* mutants was more than that in AC plants under normal conditions (Fig. 3c). Consistently, we noted that no significant change of JAM1 accumulation was detected in *atg7* mutants after the application of MG132, ConA or AZD8055 under normal conditions (Fig. 3c). Similarly, RKN-reduced JAM1 accumulation was recovered with ConA treatment, but was further decreased with AZD8055 treatment, while MG132 treatment did not change the JAM1 protein level in AC roots after RKN infection (Fig. 3d). Importantly, no significant change of JAM1 accumulation was detected in *atg7* mutants with the application of MG132, ConA or AZD8055 after RKN infection (Fig. 3d). These results reveal that JAM1 protein is degraded through the autophagy pathway rather than the proteasome pathway after RKN infection.

To further analyze the function of JAMs in tomato RKN resistance, we constructed tomato *jam1*, *jam2* and *jam1/jam2* double (*jam1/2*) mutants (Supplementary Fig. 5c–e). As shown in Fig. 3e, f, *jam* mutants all showed higher RKN resistance and fewer root knots than AC plants, and *jam1/2* mutants had the least number of root knots compared to AC, with a reduction of 56.6%. The expression levels of *PDF1.2a* and *PDF1.2b* were further upregulated in *jam* mutants compared with AC plants after RKN infection (Supplementary Fig. 6a, b). We discovered that the expression of *PDF1.2a* and *PDF1.2b* was increased by the silencing of *JAM3*, and root knot numbers were also lower in *JAM3*-

silenced plants than in TRV-control plants (Supplementary Fig. 7a–c). Thus, JAMs may negatively regulate RKN resistance in tomato. To fully determine the role of JAMs in RKN resistance, we also compared the plant growth phenotypes. As shown in Supplementary Figs. 6c and 7d, there was no significant difference in the growth phenotype between AC and mutants, TRV control and silencing plants with or without RKN infection after 5 weeks.

Interestingly, the formation of autophagosomes was significantly increased in the roots of *jam1*, *jam2*, *jam1/2* mutants and *JAM3*-silenced plants as detected by the formation of GFP-ATG8f-labeled punctate autophagosomes after RKN infection, compared with AC and TRV-control plants (Fig. 3g, h and Supplementary Fig. 7e, f). These results imply that JAMs may negatively regulate autophagy in tomato RKN response.

## JAMs disrupt ERF1-MED25 complex and ERF1-mediated defense response

MYC2 and ERFs are two core switches and play antagonistic roles in JA-mediated resistance toward different biotic stresses[53,54]. MED25, a subunit of the mediator transcriptional co-activator complex, physically interacts with MYC2 and ERFs, thereby forming a functional transcription complex to regulate JA-responsive gene expression[32,55]. In a previous study, we found that MYC2 negatively regulated tomato RKN defense[48]. Therefore, we hypothesized that JA may regulate RKN resistance through the ERF pathway. Through sequence alignment, we

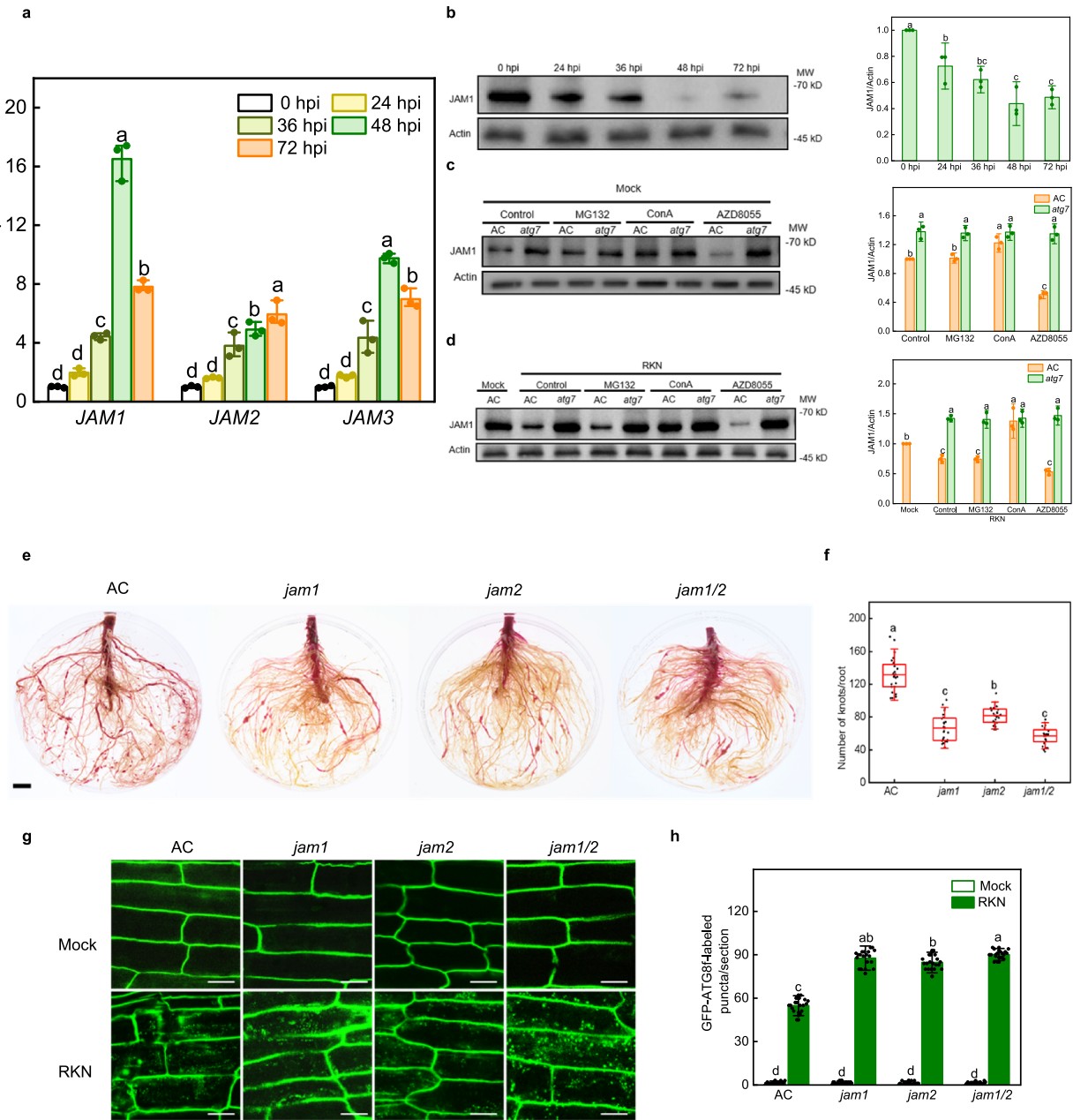

**Fig. 3 | JAM1 is degraded by the autophagy pathway and negatively regulates RKN resistance. a** Real-time quantitative PCR analysis of *JAMs* in the roots of Ailsa Craig (AC) plants at different time points after RKN infection. Total RNA was isolated from root samples collected at different time points after RKN infection. Error bars represent SD; data represent the mean ± SD (*n* = 3 biological replicates, individual dots). **b**–**d** Detection of JAM1 protein levels. **b** Changes in the protein levels of JAM1 at different time points after RKN infection. Left, western blots showing the protein levels of JAM1. Right, amounts of JAM1 determined by densitometry of protein bands from three experiments. Error bars represent SD; data represent the mean ± SD (*n* = 3 independent experiments, individual dots). **c, d** Response of WT and *atg7* mutant seedlings to the ubiquitin proteasome inhibitor MG132, the vacuolar-type ATPase inhibitor Concanamycin A (ConA) and the autophagy inducer AZD8055 with or without RKN infection. Left, western blots showing the protein levels of JAM1. Right, amounts of JAM1 determined by densitometry of protein bands from three experiments, respectively. Seedlings were pretreated with MG132 (50 μM), ConA (1 μM) or AZD8055 (5 μM) for 10 h and then were analyzed by western blotting. Band intensity was quantified by ImageJ. The ratio of JAM1/Actin in the control was set to 1. The levels of JAM1 protein were determined with anti-JAM1 antibody. The Actin protein served as loading control. Error bars represent

SD; data represent the mean ± SD (*n* = 3 independent experiments, individual dots). The experiments were repeated three times with similar results (**b**–**d**). **e** Phenotype of RKN reproduction in AC, *jam1, jam2* and *jam1/2* mutants using acid fuchsin staining after RKN infection. Bar, 1 cm. **f** The number of root knots of plants at 5 weeks after infection with RKNs. Data are presented as boxplots, with each dot representing the datapoint of one biological replicate (*n* = 25 plants). For the boxplots, the central line indicates the mean value, the bounds of the box show the 1st and 3rd quartile, and the whiskers indicate 1.5× interquartile range between the 1st and 3rd quartile. The experiments were repeated twice with similar results. **g** The direct fluorescence of GFP-ATG8f was detected in the roots by confocal fluorescence microscopic analysis with or without RKN infection (Mock). Bars, 25 μm. **h** Quantification of (**g**). The number of autophagosomes per image was quantified to calculate the autophagic activity relative to wild-type control plants, which was set to '1'. Error bars represent SD; data represent the mean ± SD (*n* = 20 samples, individual dots). The experiments were repeated twice with similar results. Different letters above bars indicate a significant difference at the *P* < 0.05 level by one-way ANOVA with Tukey's multiple comparisons test. Exact *P*-values of statistic tests are provided in the Source data file.

found that tomato ERF1 (Solyc09g089930) is a homologous protein to Arabidopsis ORA59 and ERF1 (Supplementary Fig. 8). We generated tomato *erf1* and *med25* knockout mutants by CRISPR-Cas9 system to investigate the role of ERF1 and MED25 in tomato RKN resistance (Supplementary Fig. 5f, g). As shown in Fig. 4a, b, *erf1* and *med25* mutants were more sensitive than wild-type AC after 5 weeks of RKN infection. The number of root knots in *erf1* and *med25* mutants was significantly increased by 69.4% and 64.2% compared with that of AC, respectively. RKN-induced expression of *PDF1.2a* and *PDF1.2b* was completely suppressed in the roots of *erf1* and *med25* mutants (Fig. 4c). Thus, these findings hint that ERF1 and MED25 may positively regulate tomato RKN resistance.

To explore the relationship among ERF1, MED25 and JAMs in tomato, we first performed Y2H and BiFC assays to confirm the interaction between each pair of proteins. ERF1 only interacted with MED25, and there was no interaction between JAMs and ERF1 or MED25 in vitro and in vivo (Supplementary Fig. 9). To further confirm the role of JAMs in regulating the interaction between ERF1 and MED25, we performed a yeast three-hybrid (Y3H) assay and found that MED25 interacted with ERF1 on SD-Leu/Trp/Ade/His (SD-L/T/A/H) and SD-Leu/Trp/Ade/His/Met (SD-L/T/A/H/M) media; however, the interaction between ERF1 and MED25 was severely reduced when JAM1 was co-expressed on SD-L/T/A/H/M medium (Fig. 4d). Subsequently, we carried out pull-down experiments to further test the effect of JAM1 on ERF1 and MED25 interaction. As shown in Fig. 4e, GST-ERF1 pulled down MBP-MED25, while when the amount of His-JAM1 was increased, the ability of GST-ERF1 to pull down MBP-MED25 was decreased. Moreover, we used an increasing concentration of His protein as a negative control and found that the ability of GST-ERF1 to pull down MBP-MED25 was not affected by His. In order to detect the relationship between ERF1, MED25, and JAM1 in vivo, we used Co-IP assays for validation analysis. Similar to the results of in vitro experiments, ERF1 protein was detected in the MED25 protein complex of immunoprecipitation, but after co-expression of JAM1, ERF1 protein could not be co-immunoprecipitated (Fig. 4f). Taken together, these data confirmed that JAMs interfere with ERF1 and MED25 interaction. Based on these results, we focused on whether the transcription function of ERF was affected in the subsequent study.

Next, we found that ERF binding motifs (CCGACC and ACCGAC) are present in the promoter region of *PDF1.2a* and *PDF.2b*, respectively (Supplementary Fig. 10a). To further examine whether MED25 and JAM1 are involved in the transcriptional regulation of *PDF1.2a* and *PDF1.2b* by ERF1, electrophoretic mobility shift assay (EMSA) showed that His-ERF1 recombinant proteins bound DNA probes containing the CCGACC motif of *PDF1.2a* and ACCGAC motif of *PDF1.2b*, but failed to bind with mutated probes in which CCGACC and ACCGAC motifs were replaced by AAAAAA (Fig. 4g). The ERF1-bound probe signals were increased in the presence of MBP-MED25 recombinant proteins; however, ERF1-bound probe signals decreased progressively with an increasing concentration of GST-JAM1 recombinant protein (Fig. 4g). To confirm whether ERF1 binds the promoters of *PDF1.2a* and *PDF1.2b* in vivo and the importance of JAM1 and MED25 for ERF1 in response to RKN infection, we then constructed a *JAM1* over-expressing line (*JAM1* OE#) (Supplementary Fig. 11) and subsequently transferred the *ERF1*-GFP vectors into the roots of wild-type AC, *med25* mutants and *JAM1* OE# plants by hairy root transformation[56]. Chromatin immunoprecipitation (ChIP)-real-time quantitative PCR (qPCR) analyses indicated that ERF1 directly bound to the promoters of *PDF1.2a* and *PDF1.2b* after RKN infection in vivo, whereas the binding ability of ERF1 to the promoters of *PDF1.2a* and *PDF1.2b* was impaired in both *med25* mutants and *JAM1* OE# plants. (Fig. 4h). All the above results demonstrate that MED25 acts as a co-activator of ERF1 to regulate JA-related defense genes, and JAM1 interferes with the transcriptional activity of the ERF1-MED25 complex in tomato response to RKN infection.

## RKN-induced *ERF1* expression is dependent on both JA contents and autophagy and is self-regulated

To further explore the relationship between autophagy and the JA pathway, we assessed changes in JA and JA-Ile contents in the roots of AC, *atg7*, JA biosynthetic mutant *spr2* and its wild-type Castlemart (CM). RKN induced a significant increase of JA and JA-Ile contents in AC, *atg7* and CM plants, while JA and JA-Ile contents remained at low levels in the *spr2* mutant with or without RKN infection (Fig. 5a, b). Thus, autophagy does not affect JA biosynthesis after RKN infection. Then, we detected the expression and protein accumulation of ERF1 in the roots of AC, *atg7*, CM and *spr2* at 48 hpi with RKNs. RKNs induced the expression and protein accumulation of ERF1 in the roots of two wild types; however, RKN-induced ERF1 expression and protein accumulation were completely compromised in *atg7* and *spr2* mutants (Fig. 5c, d), suggesting JA and autophagy both positively regulate ERF1 expression and protein accumulation against RKN infection.

A recent study has shown that ERF proteins can be transcriptionally auto-regulated[35]. Through analysis of the *ERF1* promoter sequence, we found that the −843 to −838 bp region of the ERF1 promoter contains a putative GCC box (Supplementary Fig. 10a). To verify whether ERF1 activates transcription from its own promoter, we performed EMSA, ChIP-qPCR and dual-luciferase assays to validate the ability of ERF1 protein to bind to its own promoter via the key GCC box. EMSA results showed that ERF1 bound to the probe from the *ERF1* promoter harboring the key GCCGCC (GCC-box) motif. This binding was successfully outcompeted by unlabeled probes but not by unlabeled mutant probes in which the GCCGCC motif was replaced by AAAAAA. ERF1 failed to bind to labeled mutant probes (Fig. 5e). Moreover, ChIP-qPCR and dual-luciferase assays, respectively, revealed that ERF1 directly bound to its own promoter (Fig. 5f) and activated its own expression (Fig. 5g).

## ERF1 positively regulates *ATGs* in response to RKN infection

To investigate the role of JA and ERF1 in RKN-induced autophagy, we examined the formation of GFP-ATG8f-labeled autophagosomes after RKN infection in the roots of wild-types, *erf1* and *spr2* mutants. RKN induced the GFP-ATG8f-labeled autophagosome accumulation in the roots of two wild types; however, the increase in RKN-induced GFP-ATG8f-labeled autophagosome number was partially impaired in *spr2* roots and completely compromised in *erf1* roots (Fig. 6a, b). Next, we compared the *ATG* expression in wild-types, *erf1* and *spr2* mutants with or without RKN infection. In the two wild types, RKN induced the expression of numerous *ATGs*, while the expression of *ATGs* was blocked in *erf1* and *spr2* mutants (Supplementary Fig. 12). Thus, ERF1 is critical for RKN-induced autophagosome formation and *ATGs* expression.

To further examine the possible regulation of *ATGs* by ERF1 during RKN infection, we inspected 3 kb of sequence upstream of the predicted *ATGs* transcriptional starting site and found the ACCGAC motif in the promoters of *ATG1a*, *ATG1b*, *ATG8d* and *ATG13b* and the CCGACC motif in the *ATG8a* promoter (Supplementary Fig. 10a). We then performed EMSA and found that recombinant His-ERF1 protein could bind to the promoters of *ATG8a*, *ATG8d* and *ATG13b*; this binding was successfully outcompeted by unlabeled DNA probes, but not by unlabeled mutant probes in which the ACCGAC and CCGACC motifs were replaced by AAAAAA (Fig. 6c). We also noted that ERF1 failed to bind with DNA probes from *ATG1a* and *ATG1b* promoters (Supplementary Fig. 10b). These results indicated that ERF1 specifically bind to the *ATG8a*, *ATG8d* and *ATG13b* promoters in vitro. To further analyze whether ERF1 regulates *ATG8a*, *ATG8d* and *ATG13b* transcription in vivo, we performed ChIP-qPCR assays on the *ERF1*-GFP plants by hairy root transformation. As shown in Fig. 6d, ERF1 can bind to the promoters of its three target genes in vivo. Meanwhile, ERF1 activated the expression of these target genes as revealed by dual-luciferase assays in transiently transformed *Nicotiana benthamiana* leaves

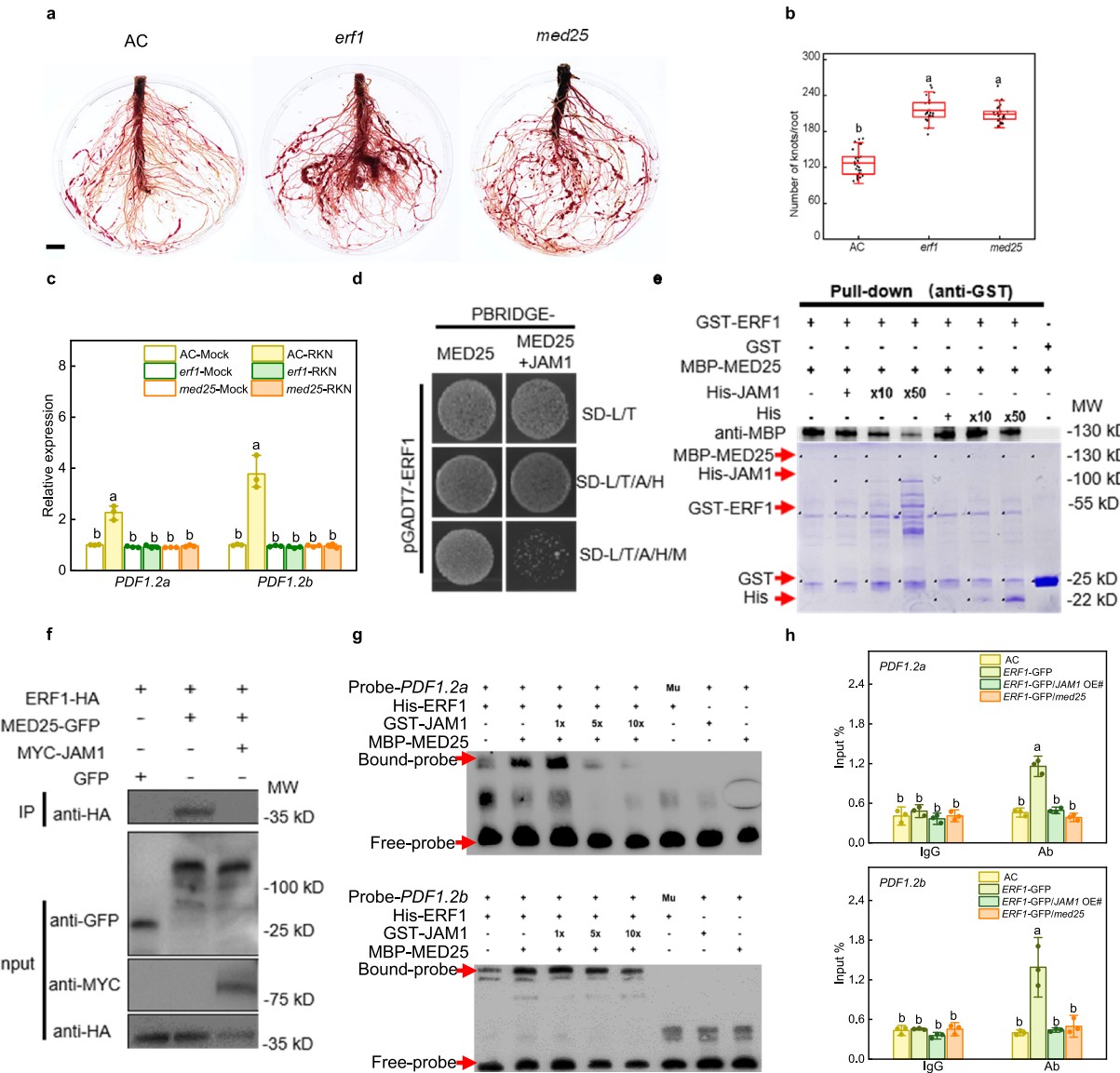

**Fig. 4 | JAM1 interferes with the binding of ERF1 to MED25 and attenuates the transcriptional regulatory capacity of ERF1. a** Phenotype of RKN reproduction in Ailsa Craig (AC), *erf1* and *med25* mutants using acid fuchsin staining 5 weeks after RKN infection. Bar, 1 cm. **b** The number of root knots of plants at 5 weeks after infection with RKNs. Data are presented as boxplots, with each dot representing the datapoint of one biological replicate (*n* = 25 plants). For the boxplots, the central line indicates the mean value, the bounds of the box show the 1st and 3rd quartile, and the whiskers indicate 1.5× interquartile range between the 1st and 3rd quartile. The experiments were repeated twice with similar results. **c** Expression of *PDF1.2a* and *PDF1.2b* after RKN infection in AC, *erf1* and *med25* mutants. Transcript levels were determined using RT-qPCR. Error bars represent SD; data represent the mean ± SD (*n* = 3 biological replicates, individual dots). **d** Yeast three-hybrid assays showing JAM1 interrupts the MED25-ERF1 interaction. Yeast cells co-transformed with pGADT7-ERF1 and pBridge-MED25 were grown on SD-Leu/Trp/Ade/His (SD-L/T/A/H) medium to assess ERF1-MED25 interaction. The co-transformed yeast cells were grown on SD-Leu/Trp/Ade/His/Met (SD-L/T/A/H/M) medium to induce the expression of JAM1. **e** In vitro pull-down assays showing JAM1 interferes with MED25-ERF1 interaction. Fixed amounts of GST-ERF1 and MBP-MED25 fusion proteins were incubated with an increasing amount of His-JAM1 fusion protein or His protein. Asterisks indicate the specific bands of recombinant proteins. Protein

samples were immunoprecipitated with anti-GST antibody and immunoblotted with anti-MBP antibody. **f** In vivo Co-IP assays showing JAM1 interferes with MED25-ERF1 interaction. Total proteins were extracted from *N. benthamiana* leaves transiently co-expressing *ERF1-HA*, *MED25-GFP*, *MYC-JAM1*, or empty *GFP* after 48 h of infiltration. Proteins were immunoprecipitated with agarose beads conjugated with GFP antibody and immunoblotted with HA antibody. **g** Electrophoretic mobility shift assays showing JAM1 interferes with, and MED25 enhances ERF1 binding to DNA probes from the *PDF1.2a* and *PDF1.2b* promoter in vitro. 5- and 10-fold excesses of GST-JAM1 protein were used for competition. Mu, mutated probes in which the CCGAC-containing motifs (5′-CCGACC-3′ and 5′-ACCGAC-3′) were replaced with 5′-AAAAAA-3′. The experiments were repeated twice with similar results (**d**–**g**). **h** Chromatin immunoprecipitation (ChIP)-real-time quantitative PCR assays showing the effect of JAM1 and MED25 on ERF1 transcriptional regulation of *PDF1.2a* and *PDF1.2b* upon RKN infection. The epitope-tagged protein-chromatin complexes were immunoprecipitated with an anti-GFP antibody (Ab). Control reactions were treated with mouse immunoglobulin G (IgG). Error bars represent SD; data represent the mean ± SD (*n* = 3 biological replicates, individual dots). Different letters above bars indicate a significant difference at the *P* < 0.05 level by one-way ANOVA with Tukey's multiple comparisons test. Exact *P*-values of statistic tests are provided in the Source data file.

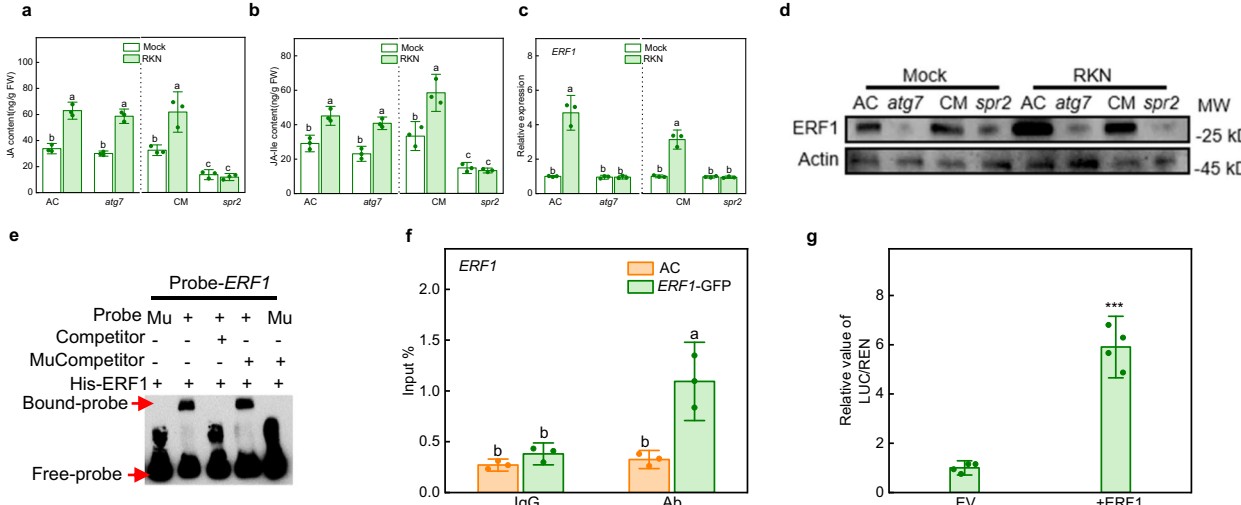

**Fig. 5 | The relationship between autophagy and JA pathway and the role of ERF1. a**, **b** Content of JA (**a**) and JA-Ile (**b**) in Ailsa Craig (AC), *atg7*, Castlemart (CM) and *spr2* mutants at 48 hpi with RKN. JA jasmonic acid, JA-Ile jasmonoyl-isoleucine. Error bars represent SD; data represent the mean ± SD (*n* = 3 biological replicates, individual dots). **c** Gene expression of *ERF1* in AC, *atg7*, CM and *spr2* mutants at 48 hpi with RKNs. Error bars represent SD; data represent the mean ± SD (*n* = 3 biological replicates, individual dots). **d** Protein levels of ERF1 in AC, *atg7*, CM and *spr2* mutants at 48 hpi with RKNs. The anti-ERF1 antibody was used to detect the protein level of ERF1 and the Actin protein was used as a loading control. The experiments were repeated twice with similar results. **e** Electrophoretic mobility shift assays to test His-ERF1 recombinant proteins bound to the probe from the *ERF1* promoter harboring the key GCCGCC (GCC-box) motif (probe-*ERF1*). Mu, mutated probe in which the GCCGCC motif was changed to AAAAAA. Competitor and mutant competitor (Mu-competitor) were used at 1000-fold. The experiments

were repeated twice with similar results. **f** Chromatin immunoprecipitation (ChIP)-real-time quantitative PCR assays to test direct binding of ERF1 to its self-promoter with RKN infection. The epitope-tagged protein-chromatin complexes were immunoprecipitated with an anti-GFP antibody (Ab). Control reactions were treated with mouse immunoglobulin G (IgG). Error bars represent SD; data represent the mean ± SD (*n* = 3 biological replicates, individual dots). **g** Regulatory effects of ERF1 on the promoter of *ERF1* as determined by dual-luciferase assays. The ratios of firefly luciferase/*Renilla* luciferase (LUC/REN) of the empty vector (EV) plus promoters under normal conditions were set as '1'. Error bars represent SD; data represent the mean ± SD (*n* = 4 biological replicates). Different letters above bars indicate a significant difference at the *P* < 0.05 level one-way ANOVA with Tukey's multiple comparisons test (**a**–**c**, **f**). The asterisks in (**g**) indicate significant difference as assessed by two-sided Student's *t*-tests; *** *P* < 0.001. Exact *P*-values of statistic tests are provided in the Source data file.

(Fig. 6e). The overall data show that ERF1 positively regulates *ATGs* expression, indicating ERF1 functions as an essential regulator responsible for autophagy.

## Discussion

Autophagy is a highly conserved vacuole-mediated degradation process of intracellular components, which not only maintains the stability of intracellular metabolism but also plays important roles in plant anti-pathogenic mechanisms. Autophagy can target the virulence protein βC1 for degradation against geminivirus Cotton leaf curl Multan virus (CLCuMuV) infection[5]. It has also been shown that autophagy negatively regulates plant resistance to pathogens. Arabidopsis *atg2* mutants exhibited powdery mildew resistance and mildew-induced cell death, revealing that autophagy suppresses cell death and defense response to the biotrophic pathogen[57]. However, the role of autophagy in plant response to nematodes has not been reported. In this study, we provided evidence that RKN infection induced the expression of a large number of *ATGs*, leading to the accumulation of autophagosomes. Observation of root phenotypes and count of root knot numbers showed that the resistance of *atg* mutants to RKNs was attenuated, and the lack of autophagy resulted in the restricted expression of *PDF1.2a* and *PDF1.2b* induced by RKNs (Fig. 1). These findings demonstrated that autophagy played a crucial role in tomato defense against RKNs.

Autophagy can be non-selective or selective[17]. Selective autophagy is particularly important in plants and mediates the degradation of target compounds through specific cargo receptors, thus playing a crucial role in responding to various environmental stresses[58]. Recent studies have also demonstrated that selective autophagy plays a key role in the elimination of invading pathogens (such as bacteria, viruses, and fungi)[9–11]. Notably, selective autophagy receptors are essential for

plant disease resistance. They recognize specific autophagy substrates on the one hand and interact with the autophagosome marker protein ATG8 on the other hand to facilitate the delivery of captured autophagic cargo to phagophores (precursors of autophagosomes) for degradation. These selective autophagy cargo receptors usually contain ATG8-interacting motifs (AIMs) or ubiquitin-interacting motifs (UIMs) for ATG8 binding. In plants, several ATG8-interacting proteins, including NBR1[21], outer membrane tryptophan-rich sensory protein (TSPO)[59], ATG8-interacting 1 (ATI1), ATG8-interacting 2 (ATI2)[60], ATG8-interacting 3 (ATI3)[61] and the intrinsic 26S proteasome base subunits, regulatory particle non-ATPase 10 (RPN10) (ref. [62]) have been identified and found to mediate selective autophagy of specific cellular components during plant responses to different environmental stresses. In this study, we found that JAM1 interacted with ATG8a through Y2H screens and verified that JAM1 and its homologous proteins (JAM2 and JAM3) interacted with ATG8s through the N-terminal AIM domain (Fig. 2 and Supplementary Figs. 2 and 4). In line with currently considered selective receptors such as TSPO[59], JAM proteins are mainly degraded by autophagy after nematode infestation. However, unlike selective receptors such as NBR1 that mediate selective autophagy and promote the aggregation and subsequent targeting of ubiquitylated protein cargoes to ATG8-positive autophagosomes under stresses[21], JAM1 is degraded by autophagy and *jams* mutants show increased formation of autophagosomes and higher resistance after nematode infection. Therefore, JAMs serve as the substrates of selective autophagy in tomato after nematode infection rather than cargo receptors responsible for the degradation of other substrates.

JAMs act as transcriptional repressors and negatively regulate JA signaling to play a key role in regulating JA-mediated stress responses and various metabolic pathways[63]. Recently, JAM1-JAM3 has been termed MYC2-targeted bHLH1 (MTB1)-MTB3, which negatively

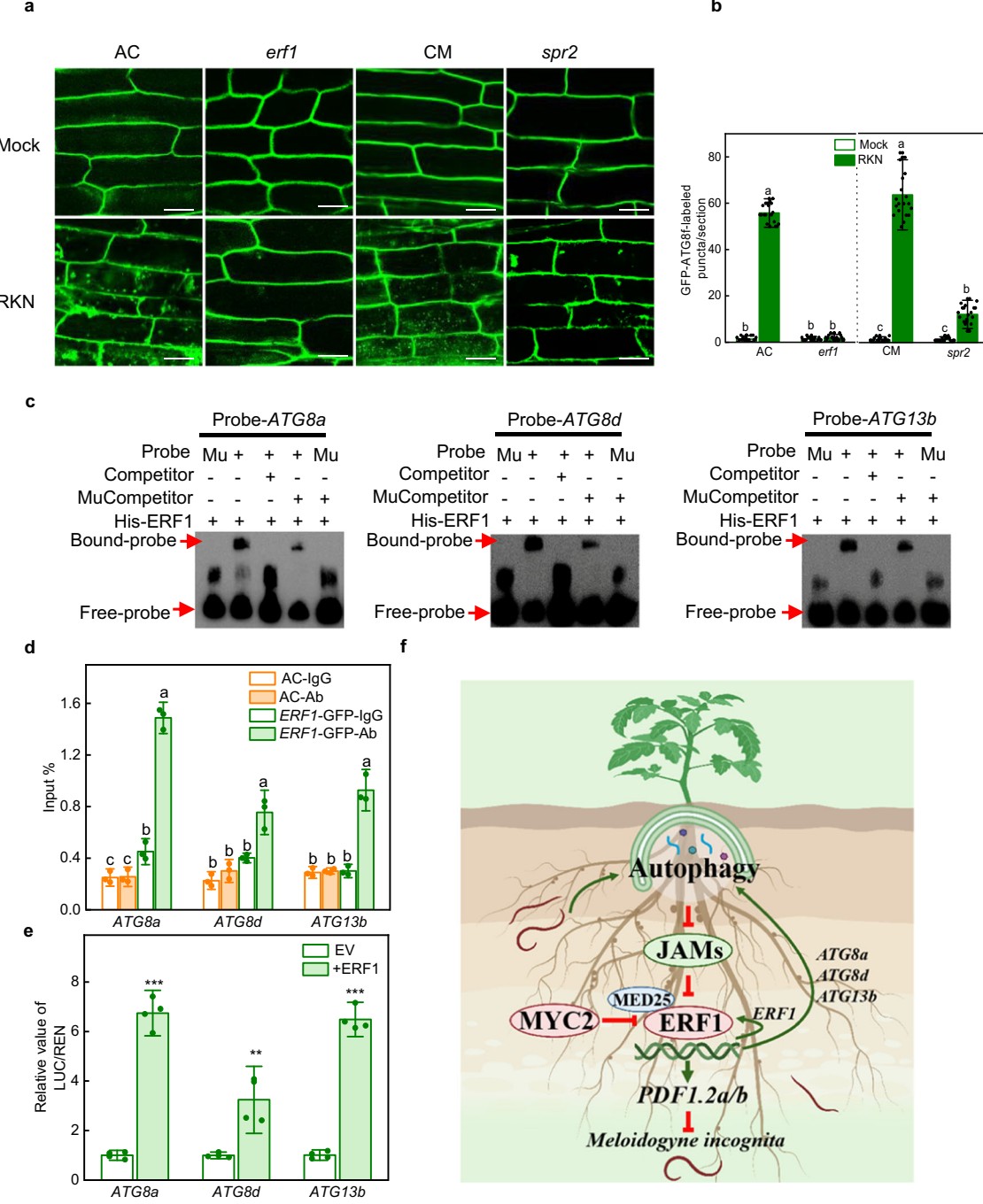

regulates MYC2-mediated JA signaling and plant resistance to *Helicoverpa armigera* larvae attack[64]. In our study, we found that RKNs induced the expression of *JAMs*, but their protein levels significantly decreased after RKN infection (Fig. 3a–d). The phenotype and expression of resistance genes in *jam* mutants showed that JAMs negatively regulated nematode resistance (Fig. 3e, f and Supplementary Figs. 6 and 7). In addition, *jam* mutants hardly affect plant growth (Supplementary Figs. 6 and 7), indicating that knocking out of *JAMs* is an effective means to improve nematode resistance in tomato.

MYC2 and ERF transcription factors are two antagonistic branches in JA-mediated resistance toward different biotic stresses in plants. In general, the MYC branch is associated with wound response and defense against insect herbivores, whereas the ERF branch is associated with enhanced resistance to necrotrophic pathogens[65]. Our previous study showed that MYC2 negatively regulated RKN resistance

in tomato[48]. Here, we showed that ERF1 was a key regulator for tomato RKN resistance, and it required transcriptional co-activator MED25 to improve its binding capacity, thereby regulating the expression of RKN resistance genes. Although JAMs do not interact with ERF1 and MED25 alone, they regulate the interaction between ERF1 and MED25 and interfere with the transcriptional ability of ERF1 (Fig. 4 and Supplementary Fig. 9). Previous studies have also shown that JAMs (also called MTBs) interfere with the MYC2-mediated JA signaling pathway in Arabidopsis and tomato, thereby negatively regulating plant resistance[63,64]. Thus, JAMs are key nodes of both branches of the JA signaling pathway.

In recent years, the relationship between autophagy and phytohormones has been reported. For example, Brassinosteroid (BR) regulates *ATGs* under nitrogen deficiency through the downstream transcription factor BRASSINAZOLE RESISTANT1 (BZR1)[4], while

**Fig. 6 | JA-mediated autophagy is partially dependent on ERF1. a** The direct fluorescence of GFP-ATG8f was detected in the roots of Ailsa Craig (AC), *erf1*, Castlemart (CM) and *spr2* plants by confocal fluorescence microscopic analysis with or without RKN infection. Bars, 25 μm. **b** Quantification of (**a**). The number of autophagosomes per image was quantified to calculate the autophagic activity relative to wild-type control plants, which was set to '1'. Error bars represent SD; data represent the mean ± SD (*n* = 20 samples, individual dots). The experiments were repeated twice with similar results. **c** Electrophoretic mobility shift assays for the combination of ERF1 and *ATG8a* promoter with CCGACC motif (probe-*ATG8a*), both *ATG8d* and *ATG13b* promoters with ACCGAC motifs (probe-*ATG8d*/probe-*ATG13b*). Mu, mutated probes in which the CCGACC and ACCGAC motifs were changed to AAAAAA. Competitor and mutant competitor (Mu-competitor) were used at 1000-fold. The experiments were repeated twice with similar results. **d** Chromatin immunoprecipitation (ChIP)-real-time quantitative PCR assays for direct binding of ERF1 to *ATG8a*, *ATG8d* and *ATG13b* with RKN infection in vivo. The epitope-tagged protein-chromatin complexes were immunoprecipitated with an anti-GFP antibody (Ab). Control reactions were treated with mouse immunoglobulin G (IgG). Error bars represent SD; data represent the mean ± SD (*n* = 3 biological replicates, individual dots). **e** Dual-luciferase assays for the regulatory effects of ERF1 on the promoters of *ATG8a*, *ATG8d* and *ATG13b*. The ratios of firefly luciferase/Renilla luciferase (LUC/REN) of the empty vector (EV) plus promoters under normal conditions were set as '1'. Error bars represent SD; data represent the mean ± SD (*n* = 4 biological replicates, individual dots). Different letters above bars indicate a significant difference at the *P* < 0.05 level by one-way ANOVA with Tukey's multiple comparisons test (**b**, **d**). The asterisks in (**e**) indicate significant difference as assessed by two-sided Student's *t*-tests; ** *P* < 0.01, *** *P* < 0.001. Exact *P*-values of statistic tests are provided in the Source data file. **f** Proposed mechanism reveals that autophagy promotes JA-mediated defense against RKNs by forming a positive feedback loop to degrade JAMs and activate ERF1 transcriptional activity. JAMs, the components of the JA pathway, interfere with the binding of ERF1 and MED25 and affect the ability of ERF1 to activate autophagy, defense genes (*PDF1.2a* and *PDF1.2b*) and transcriptional self-regulation. In addition, the MYC2 branch may also interfere with the MED25-ERF1 complex to negatively regulate RKN resistance. The red blocking symbols indicate inhibition, green arrows indicate promotion. This graphic was created with BioRender.com.

---

autophagy negatively regulates BR signaling under drought and starvation through the selective receptor, DOMINANT SUPPRESSOR OF KAR 2 (DSK2)[22]. Furthermore, autophagy can also participate in plant defense through hormonal pathways. Over-expression of apple *MdATG18a* improved resistance to *Diplocarpon mali* infection by increasing SA levels, suggesting that autophagy may play an active role in regulating SA accumulation[66]. In contrast, high concentrations of SA lead to autophagy-mediated senescence and programmed cell death[38]. However, little has been reported on the interaction between autophagy and the JA signaling pathway. Using autophagy mutant *atg7* and JA biosynthetic mutant *spr2*, we have discovered that autophagy mutation hardly affects JA synthesis after nematode infection but affects ERF1 expression and protein content (Fig. 5a–d), while the JA biosynthetic mutation *spr2* results in compromised expression of key *ATGs* and autophagosome formation after RKN infection (Fig. 6a, b and Supplementary Fig. 12), suggesting possible crosstalk between autophagy and JA signaling but not JA synthesis. ERF1 serves as a key node for autophagy-JA crosstalk. On the one hand, autophagy activates ERF1 transcriptional activity by degrading JAMs; simultaneously, ERF1 can achieve transcriptional auto-regulation and further regulate *ATGs* expression and the formation of autophagosomes.

In our study, we revealed the mechanism by which autophagy regulated tomato RKN resistance through promoting the JA signaling pathway and elucidated that selective autophagy can degrade the JA signaling negative regulators JAMs to activate the JA-ERF signaling branch, suggesting that autophagy may play complex but important roles in plant environmental responses. In future studies, it will be interesting to investigate whether autophagy is involved in other regulatory mechanisms of JA signaling. In addition, the discovery of the functions of autophagy in plant-nematode interactions provides another interesting direction for future exploration. On the one hand, plant autophagy may reduce nematode infection by degrading nematode secretion effector proteins; on the other hand, nematode secretion effector proteins may trigger plant immunity to initiate autophagy while also having the possibility to interfere with autophagy activity.

Taken together, our study demonstrated that autophagy positively regulated JA signaling and RKN resistance in tomato plants (Fig. 6f). In brief, autophagy promotes JA-mediated defense against RKNs by forming a positive feedback loop to degrade JAMs and stimulates the JA-ERF1 branch. ERF1 functions as a transcriptional regulator for JA-responsive genes. Furthermore, the expression of *ATGs* is regulated by JA signaling in an ERF1-dependent manner, and *ERF1* expression is also self-regulated. In addition, the MYC2 branch may also interfere with the MED25-ERF1 complex to negatively regulate RKN resistance. Our results provide new insight into the JA-mediated resistance mechanism and potential approaches to enhance plant nematode resistance by manipulating the autophagy pathway.

## Methods

### Plant materials and growth conditions
Tomato (*Solanum lycopersicum*) cultivars Ailsa Craig (AC) and Castlemart (CM), which are both RKN susceptible genotypes, were used in all experiments. JA-deficient mutants (*spr2*) were in the CM background, and other mutants and transgenic lines were in AC background. Besides the mutants and transgenic lines generated in this study, *atg6*, *atg10* and *spr2* were used[56,67]. Tomato seedlings were cultivated in sterilized sandy loam soil and irrigated with 1/2 Hoagland's nutrient solution. Plants were grown at 20–23 °C with a 14-h light/10-h dark cycle and 600 μmol m⁻² s⁻¹ light intensity in a growth room.

### *Meloidogyne incognita* infection assays
RKN populations were maintained on susceptible tomato plants before being used for plant infection. Plants were infected with RKNs according to the method described by Wang et al.[68]. The tomato plants were inoculated with 1000 second-stage juveniles (J2s) or mock-inoculated with water over the surface of the sand around the primary roots using a pipette.

### RKN staining
The staining method was as previously described[69]. Plants were collected at 35 days post-inoculation (dpi) and stained with acid fuchsin solution (0.35% acid fuchsin, 25% acetic acid) to visualize root knots. To examine RKN colonization, more than 20 roots of each treatment were placed in acidified glycerin and photographed. More than 20 seedlings of each genotype were used for the determination of the knots on individual root systems in each experiment.

### Constructs and plant transformation
Gene loss- and gain-function stable tomato lines were generated through gene editing and over-expression approaches, respectively. Constructs for CRISPR/Cas9 mutagenesis were generated as previously described[56]. Briefly, sgRNA targeting sequences were designed using the CRISPR-P server (http://cbi.hzau.edu.cn/cgi-bin/CRISPR). All sgRNA sequences are listed in Supplementary Table 1. The synthesized sequences were annealed and inserted into the *BbsI* site of AtU6-sgRNA-AtUBQ-Cas9 vector, and the AtU6-sgRNA-AtUBQ-Cas9 cassette was inserted into the *HindIII* and *KpnI* sites of the pCAMBIA1301 binary vector. All resulting plasmids above were transformed into *Agrobacterium tumefaciens* strain EHA105 and infected into AC cotyledons. Transformed plants were selected on the basis of hygromycin resistance, and knockouts were identified by sequencing PCR amplicons of

the target loci. After confirmation with Sanger sequencing (Supplementary Fig. 5), independent homozygous Cas9-free F2 lines from each gene-edited mutation were used for the study.

To generate *JAM1* over-expression lines, *JAM1* full-length coding sequence (CDS) was cloned into the PFGC1008-HA vector under the control of the cauliflower mosaic virus (CaMV) *3SS* promoter. Transgenic plants were generated by *A. tumefaciens*-mediated transformation. Stable expression of transgenes was confirmed by immunoblotting using the anti-HA antibody (Abcam, ab187915) (Supplementary Fig. 11). Independent homozygous T2 line was used in this study. The primers used for plasmid construction are listed in Supplementary Table 2.

For the generation of tomato with transgenic roots, the *3SS::GFP-ATG8f* and *3SS::ERF1*-GFP vectors were constructed as described previously[56]. Briefly, the resulting plasmids were transformed into *A. rhizogenes* strain K599 (Tolobio, CC96315), which was used to infect the hypocotyls of 6-day-old aseptic tomato seedlings in the tissue culture bottles filled with 1/2 Murashige & Skoog (MS) medium (PhytoTechnology, M519). The infected seedlings were kept in 1/2 MS medium for 2 weeks until the hairy roots were generated from the wounded sites. The original roots were cut, and the seedlings were moved to sandy loam soil for further treatment. Primers used for plasmid construction are listed in Supplementary Table 2.

The tobacco rattle virus (TRV)-based virus-induced gene silencing (VIGS) was used for silencing of *JAM3* gene in AC seedlings. 300-bp *JAM3* cDNA fragment for VIGS was generated by PCR amplification using specific primers (Supplementary Table 2). The amplified fragment was digested with *EcoRI* and *BamHI* and ligated into the same sites of TRV2. The resulting plasmid was transformed into *A. tumefaciens* strain GV3101. *A. tumefaciens*-mediated virus infection was performed as described previously[70]. VIGS seedlings which showed about 40% transcript levels of control plants were used (Supplementary Fig. 7g).

### RNA isolation and RT-qPCR analysis
Total RNA was extracted from tomato roots using an RNA extraction kit (Tiangen, DP419) and reverse-transcribed using a ReverTra Ace qPCR RT kit (Vazyme, R223) according to the manufacturer's instructions. RT-qPCR was performed using a Light Cycler 480 II detection system (Roche) with a SYBR Green PCR Master Mix Kit (Vazyme, Q711). The relative gene expression was calculated using the previously described method with *Actin* and *Ubiquitin3* as internal controls for normalization[71]. The heatmap analysis was performed using MeV version 4.9 (http://www.mev.tm4.org/). The intensity of the color bar at the top indicates the levels of expression. Gene-specific primers used for RT-qPCR are presented in Supplementary Table 3.

### Protein extraction and western blotting
Protein extraction from root samples and western blotting were performed as described previously[4]. For ATG8 and ATG8-PE detection, the denatured proteins were separated on a 13.5% SDS-PAGE gel in the presence of 6 M urea. For western blotting, proteins were separated by 10% SDS-PAGE gel and then transferred to nitrocellulose membranes. The HA-tagged JAM1 protein was detected with anti-HA monoclonal antibody (Abcam, ab187915); Actin protein was detected with anti-Actin polyclonal antibody (Abcam, ab197345); ATG8 protein was detected with anti-ATG8 polyclonal antibody (Agrisera, AS142769); JAM1 protein and ERF1 protein were detected with anti-JAM1 polyclonal antibody (Biospring, BMH210101) and anti-ERF1 polyclonal antibody (Biospring, BMH210102), respectively.

### Autophagosome detection
Confocal detection of autophagy in tomato roots was described previously[56]. Tomato roots over-expressing *GFP-ATG8f* were cut into small sections, and autophagosomes were observed with a Nikon

A1plus confocal microscope (Nikon) with excitation at 488 nm and emission at 493 to 558 nm. ConA (1 μM) was done for 3 h prior to confocal microscopy. For each treatment, 20 representative pictures were taken, and the number of autophagosomes in each image was counted.

### In vivo degradation assays
Plants were treated with minor modifications as described by Nolan et al.[22]. Plants were soaked in 0.5X Linsmaier and Skoog (LS) liquid medium containing DMSO, 50 μM MG132 (Sigma, M7449), 5 μM AZD8055 (Macklin, A837130) or 1 μM ConA (Glpbio, GC17519). Plants were vacuum infiltrated with the different chemicals for 5 min and then incubated at room temperature for 10 h before western blotting analysis. The JAM1 levels were measured with an anti-JAM1 antibody. Actin was used to demonstrate equivalent protein loading.

### Subcellular localization
JAM1, JAM2 and JAM3 were each cloned into a pCAMBIA2300 (CAMBIA) vector with a GFP tag at the C-terminus under the control of the *3SS* CaMV promoter. Transgenic tobacco leaves were infiltrated with the resulting 3*3SS::JAMs* (*JAM1*, *JAM2*, *JAM3*)-GFP constructs mediated by *A. tumefaciens strain* GV3101. At 48 h after infiltration, the fluorescence of the leaves was observed and recorded with a Zeiss LSM 780 confocal microscope; the excitation/emission wavelengths for GFP were 488 nm/500–530 nm and 561 nm/580–620 nm for NLS-mCherry[72].

### Y2H assays
The tomato cDNA library construction and Y2H screening were performed following the manufacturer's protocol (Clontech). The coding sequence of ATG8a was cloned into the pGBKT7 vector as bait and transformed into the AH109 yeast strain. For Y2H screening, colonies were directly cultured onto SD-Leu/Trp/Ade/His (SD-L/T/A/H) plates after mating. For further confirmation of putative positive clones, the mated colonies carrying pGBKT7-ATG8s and pGADT7-JAMs including JAMs mutated AIM motifs (JAMs^mAIM1 and JAMs^mAIM2) were transferred from SD-L/T plates to SD-L/T/A/H plates at different dilutions. Primers used for plasmid construction are listed in Supplementary Table 2.

### Y3H assays
Y3H assays were performed as described previously[64]. To construct pBridge-MED25-JAM1, MED25 CDS was cloned into the multiple cloning site (MCS) I of pBridge vector (Clontech) fused to the GAL4 BD domain, and JAM1 CDS was cloned into MCS II of the pBridge vector and expressed as the "bridge" protein only in the absence of Met. Constructs used for testing protein-protein interactions were co-transformed into yeast strain AH109. The presence of transgenes was confirmed by growing the yeast cells on the SD-L/T medium. Transformed yeast cells were spread on plates containing SD-L/T/A/H medium to assess the ERF1-MED25 interaction without the expression of JAM1 and on plates containing SD-L/T/A/H/M medium to induce JAM1 expression. Interactions were observed after 3 days of incubation at 28 °C. Primers used for plasmid construction are listed in Supplementary Table 2.

### BiFC assays
For BiFC assays, cYFP-ATG8a, JAMs-nYFP, JAMs mutated AIM1 motif (JAMs^mAIM1), ERF1-nYFP and MED25-nYFP/-cYFP were constructed using specific primers (Supplementary Table 2). Meanwhile, mCherry-ATG8f was co-expressed as autophagosome location marker[56]. *A. tumefaciens*-mediated transient expression in *N. benthamiana* leaves was performed as described[61]. After 48 h infiltration, subcellular localization of YFP or mCherry signals in leaves was determined with a Zeiss LSM 780 confocal microscope; excitation/emission wavelengths were 514 nm/520 to 560 nm for YFP and 561 nm/580 to 620 nm for mCherry.

To detect autophagy, AZD8055 (15 μM) treatment for 3 h was performed prior to confocal microscopy.

## Recombinant protein expression assays

To produce MBP-MED25 fusion protein, full-length MED25 CDS was PCR amplified and cloned into pMAL-c2X. To produce GST-ERF1 and GST-JAM1 fusion proteins, full-length ERF1 and JAM1 CDS were PCR amplified and cloned into pGEX-4T-3. To produce His-JAM1 and His-ERF1 fusion proteins, the full-length CDS of JAM1 was PCR amplified and cloned into pET-32a, while the full-length CDS of ERF1 was PCR amplified and cloned into pET-28a. All recombinant plasmids and empty plasmids (GST-tag, His-tag) were transformed into *Escherichia coli* BL21 (DE3) cells and induced by 0.5 mM isopropyl β-D-1-thiogalactopyranoside (IPTG, SIGMA, 092M4001V). The MBP-tagged and GST-tagged fusion proteins were expressed and then purified using the amylose resin (NEB, E8032LVIAL) and GST Bind Resin (Millipore, EM70541-5), respectively. The His-tagged fusion proteins were purified according to the instructions provided with the Novagen pET purification system[64].

## Pull-down assays

To detect the effect of JAM1 on ERF1-MED25 interaction in vitro pull-down assays, GST-ERF1, MBP-MED25 and His-JAM1 fusion proteins were affinity purified according to previous methods[73]. To each reaction was added 1 μg purified GST-ERF1 and MBP-MED25 proteins, with increasing amounts of His-JAM1 fusion proteins or His proteins. All the reactions were added to 1 ml reaction buffer (25 mM Tris-HCl pH 7.5, 100 mM NaCl, 1 mM DTT, and 1x protease inhibitor cocktail) at 4 °C for 2 h. Subsequently, beads were collected and washed three times with washing buffer (25 mM Tris-HCl pH 7.5, 150 mM NaCl, and 1 mM DTT). After washing, samples were denatured using SDS loading buffer and separated using SDS-PAGE. The MBP-MED25 fusion proteins were detected by immunoblotting with anti-MBP antibody (NEB, E8032LVIAL). Purified GST was used as a negative control. Gel staining was performed with Coomassie Blue (eStain™ L1 Protein Staining System, GenScript) to detect various proteins in each reaction.

## EMSA assays

To detect the role of JAM1 and MED25 in ERF1 transcriptional regulation of *PDF1.2a* and *PDF1.2b*, His-ERF1, GST-JAM1 and MBP-MED25 fusion proteins were affinity purified. EMSA was performed as previously described[64]. Briefly, the DNA probes (Supplementary Table 4) were biotin end-labeled referring to the instructions of the Biotin 3′ End DNA Labeling Kit (Pierce, 89818) and then annealed to double-stranded probe DNA. Protein-DNA complexes were analyzed according to the instructions of the Light Shift Chemiluminescent EMSA kit (Thermo Fisher, 20148). Meanwhile, increasing amounts of His-JAM1 and MBP-MED25 fusion proteins were added to the reactions. To detect the transcriptional regulation of ERF1 on *ATGs* and itself, biotin-labeled probes with or without competitors or mutant competitors (1000-fold) were incubated with His-ERF1 proteins at room temperature for 15 min, and free and bound probes were separated via a 6% non-denaturing polyacrylamide gel.

## Co-IP assays

For Co-IP assays, MYC-ATG8a and empty GFP, JAM1-GFP, or JAM1-GFP mutated AIM1 motif (JAM1^mAIM1-GFP) were co-expressed in *N. benthamiana* leaves according to Hu et al.[73]. The leaf samples were collected at 2 days post-inoculation, and ground in IP buffer (50 mM Tris-HCl, pH 7.5, 150 mM NaCl, 5 mM EDTA, 0.5% Triton X-100; 1x protease inhibitor cocktail, 2.5 μl 0.4 M DTT, 2 μl 1 M NaF, and 2 μl 1 M Na₃VO₄). Each set of GFP-tagged soluble protein immunoprecipitation was performed in 1 ml IP buffer with 15 μl of anti-GFP agarose beads (Chromotek). After 3 h of gentle shaking at 4 °C, the agarose beads were washed three times with washing buffer (150 mM NaCl, 50 mM

Tris-HCl, pH 7.5, 5 mM EDTA, and 0.1 % Triton X-100), and once more with 50 mM Tris-HCl, pH 7.5. Then, the immunoprecipitated proteins were detected by immunoblotting with indicated antibodies.

## ChIP-qPCR

ChIP experiments were performed using the EpiQuik Plant ChIP Kit (Epigentek, P-2014) according to the manufacturer's instructions[65]. Then, 2 g of wild-type or *ERF1-GFP* transgenic root tissues were harvested. Chromatin was immunoprecipitated with anti-GFP antibody (Abcam, ab290). A goat anti-mouse immunoglobulin G (IgG) antibody (EMD, Millipore AP124P) was used as the negative control. ChIP-qPCR was performed with specific primers for different promoters (Supplementary Table 5).

## Dual-luciferase transcription activity assays

The dual-luciferase assays were performed as previously described[70]. Briefly, full-length sequences of *ERF1* and the gene promoters were inserted into the pGreen II 0029 62-SK vector (SK) and pGreen II 0800-LUC vectors, respectively. The primers used for vector construction are listed in Supplementary Table 2. The promoter PCR products were cloned into the pGreenII 0800-LUC vector to induce the firefly luciferase (*LUC*) reporter gene, while the internal control *Renilla* luciferase (*REN*) reporter gene was driven by the *35S* promoter. Activity analysis of promoters was performed in *N. benthamiana* using *A. tumefaciens*-mediated transient expression. All constructs were transformed into *A. tumefaciens* strain GV3101. The *A. tumefaciens* mixtures of ERF1 and promoter with a ratio of 10:1, which were both adjusted to an $OD_{600}$ of 0.8 with infiltration buffer, were then infiltrated into the leaves of *N. benthamiana*. After infiltration for 3 days, LUC and REN activities were assayed using the Dual-LUC Reporter Gene Assay Kit (Beyotime, RG027). The relative LUC/REN activity of the combination of empty SK vectors mixed with the promoters was set at a value of 1, and the analyses were performed with three replicates.

## JA and JA-Ile quantification

The phytohormones JA and JA-Ile were extracted from tomato roots as described previously[68]. Briefly, 100 mg of frozen root material was homogenized in 1 mL of ethyl acetate, which had been spiked with D6-JA (OlChemIm, Czechoslovakia) and D6-JA-Ile (Quality control chemicals, USA) as internal standards. The samples were shaken at 200 rpm in the dark at 4 °C overnight and then centrifuged at 18,000×*g* for 10 min at 4 °C. The pellet was re-extracted with 1 mL of ethyl acetate. The supernatants were combined and evaporated to dryness under N₂. The residue was re-suspended in 0.6 mL of 70% methanol (v/v) and centrifuged. The final supernatants were pipetted into glass vials and then analyzed in a liquid chromatography tandem mass spectrometry system (Varian 320-MS LC/MS, Agilent). LC analysis was performed using an Agilent Zorbax XDB C18 column. Water with 0.05% formic acid (solvent A) and methanol (solvent B) was used as the mobile phase. The gradient program is as follows: 0–1.5 min, A: B at 60: 40; followed by 6.5 min solvent A: B at 0: 100; subsequently returning to solvent A: B to 60: 40 for 5 min, with a flow rate of 0.3 mL min⁻¹. The column temperature was kept at 40 °C, and the injection volume was 20 μL. A negative electrospray ionization mode was used for detection. The JAs were detected in MRM mode by monitoring the transitions 209.1 > 59.1 for JA; 214.3 > 62.1 for D6-JA; 322.0 > 130.0 for JA-Ile; 328.5 > 130.1 for D6-JA-Ile. Each treatment was biologically replicated three times.

## Statistical analysis

The values are represented as the mean ± SD. The difference between the two groups was assessed by Student's *t*-tests. Statistically significant differences among multiple groups were evaluated by one-way

ANOVA followed by a Tukey's test. Details of each statistical test are indicated in the figure legends.

## Reporting summary

Further information on research design is available in the Nature Portfolio Reporting Summary linked to this article.

## Data availability

All data generated or analyzed during this study are included in the main text and Supplementary Information. All tomato genes involved in this study can be found at the Sol genomics network (http://solgenomics.net/), with the following accession numbers: *ATG1a* (Solyc09g011320), *ATG1b* (Solyc10g084930), *ATG2* (Solyc01g108160), *ATG3* (Solyc06g034160), *ATG4* (Solyc01g006230), *ATG5* (Solyc02g036380), *ATG6* (Solyc05g050390), *ATG7* (Solyc11g068930), *ATG8a* (Solyc07g064680), *ATG8b* (Solyc02g080590), *ATG8c* (Solyc03g031650), *ATG8d* (Solyc10g006270), *ATG8e* (Solyc08g007400), *ATG8f* (Solyc08g078820), *ATG8h* (Solyc01g068060), *ATG9* (Solyc04g008630), *ATG10* (Solyc09g047840), *ATG12* (Solyc12g049310), *ATG13a* (Solyc03g096790), *ATG13b* (Solyc06g072980), *ATG18a* (Solyc08g006010), *ATG18b* (Solyc07g006120), *ATG18c* (Solyc01g099400), *ATG18f* (Solyc12g005230), *ERF1* (Solyc09g089930), *JAM1* (Solyc01g096050), *JAM2* (Solyc05g050560), *JAM3* (Solyc06g083980), *MED25* (Solyc12g070100), *PDF1.2a* (Solyc07g006380), *PDF1.2b* (Solyc11g028070). *Arabidopsis* genes involved in this study can be found at TAIR (www.arabidopsis.org), with the following accession numbers: *ERF1* (AT3G23240), *JAM1* (AT2G46510), *JAM2* (AT1G01260), *JAM3* (AT4G16430), *ORA59* (AT1G06160). Source data are provided with this paper.

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

## Acknowledgements

This work was supported by the National Key Research and Development Program of China (2022YFD1200502), the National Natural Science Foundation of China (32272790, 31922078 and 31872089), the Starry Night Science Fund of Zhejiang University Shanghai Institute for Advanced Study (SN-ZJU-SIAS-0011) and the Fundamental Research Funds for the Central Universities (226-2022-00122) to J.Zhou and partially by the National Natural Science Foundation of China (32020103013) to J.Y. Work in the lab of D.C.B is supported by National Science Foundation grant # MCB-2040582. The authors are grateful to S. Wang for *spr2* mutant and its wild-type Castlemart seeds.

## Author contributions

J.Zou, J.Y. and J.Zhou conceived and designed the experiments. J.Zou, X.C., C.L., M.G., Z.Q., P.Y. and G.W. performed the experiments. J.Zou, X.C., M.K.K. and J.Zhou analyzed the data. Y.B. and D.C.B. provided the critical discussion. J.Zou and J.Zhou wrote the article. All authors reviewed and revised the article.

## Competing interests

The authors declare no competing interests.
