## [Peer Review File · Nature Communications]

Autophagy promotes jasmonate-mediated defense against nematodesReviewer #1 (Remarks to the Author):

Reviewer evaluation of Zou et al.

The paper by Zou et al. describes a comprehensive analysis of the role of autophagy in JA-mediated plant defence against RKN in tomato. The authors have done a lot of work, using a various set of techniques and provide compelling evidence for the role of autophagy-mediated JAM degradation after RKN infection, mediated by interaction of JAMs with ATG8. They also nicely show how JAMs impair the ERF1-based activation of the JA signaling pathway that leads to plant defence responses against RKN. The work is novel and timely, and brings very interesting new insights for the field.

There are a few experimental elements and explanations missing. Below are my comments for improvement of the manuscript.

- 1/ The authors show a role for the JA-ERF1 branch in plant defence against RKN in tomato. However, this branch is typically associated with resistance to necrotrophic pathogens, while RKN are extremely biotrophic. How to reconcile this? Can this be discussed?
- 2/ Line 120 is rather confusing. The authors state here that JA signaling is required for tomato susceptibility to RKNs. In the rest of their manuscript, however, they always link JA with resistance, not susceptibility. See for example a few lines further (line 126) where they state that JA has a 'crucial position in RKN resistance'. This should be explained better in the text.
- 3/ It is not well-explained why the authors monitor expression of Pdf1.2a and Pdf1.2b in tomato after RKN infection. Why these genes? What does induction of these genes indicate? Are they used here as JA-marker genes? What is their role in tomato resistance against RKN?
- 4/ Susceptibility to RKN should be monitored in an atg8-mutant. Now, four mutants in other atg-genes are included. How were these 4 selected?
- 5/ the authors studied the interacting proteins of ATG8a using Y2H. This was however only done for interaction with plant proteins. Why not for nematode proteins? It would be extremely interesting to find out if RKN secretes effectors that potentially interact with ATG8s.
- 6/ The use of DTT, as inducer of autophagy, should be used as positive control for the experiments shown in Fig. 1b and Fig. 1d.
- 7/ Lines 225-238: why were these experiments done on the atg7 mutant? Why not on other atg mutants? Why was this mutant selected?
- 8/ To demonstrate that JAM1 is regulated by the autophagic pathway, the authors do assays with chemical inhibitors. An additional experiment should also be done in nematode-infected plants, using the same chemicals.
- 9/ Lines 273-276: the information provided here is based on Arabidopsis work. Is a one-to-one translation to tomato valid? Do the 2 JA-branches (MYC2 and ERF1) function similar in tomato as in Arabidopsis? Do the 2 branches differ in their role towards RKN? Apparently they do, but this should be better explained in the manuscript.
- 10/ How similar are the JAMs to MYC2?
- 11/ The susceptibility of the spr2 mutant to RKN should be presented in the manuscript.
- 12/ Line 390: this sentence seems to suggest that ERF1 only has 3 target genes. This is however not known. The authors should monitor the expression of ATG8a, ATG8b and ATG13 and other potential target genes (such as pdf1.2a & b) in the erf1-mutant using qRT-PCR. Alternatively, and RNA-seq study on this mutant could provide insights about potential other target genes of ERF1.
- 13/ What is known about the function of the various homologues of ATG8?
- 14/ Line 452: the 'inconsistency' reported here is not so surprising. The proteins are degraded by autophagy, not the mRNA.
- 15/ Discussion: Can the authors comment on the potential of interference by the nematode in autophagy? For example, by secreting effectors that interact with ATGs.
- 16/ The temperature at which the infections were done (20-23°C) are very low for tomato and RKN. Is this correct?
- 17/ The hormone measurements were done on only 3 biological replicates. The SD of these results, shown in fig. 5a, cannot be correct. Hormone levels typically vary a lot between plants, hence the error bars are expected to be larger.
- 18/ The model shown in Fig. 6f seems incomplete. Negative interactions should be better indicated. Also the knowledge on MYC2 (based on a previous publication from this group) could be

added to provide a better insight into the role of the two JA branches. What are the red and green arrows indicating?

Minor issues

- Line 51: write N-protein in full
- Line 200: ATG8a homologous proteins: reference needed
- Line 374: ERF1 is critical for ATG (not JA)-mediated autophagy.
- Line 495: 'can precisely regulate JA signaling'. This statement is too vague

Reviewer #2 (Remarks to the Author):

The present manuscript by Zou et al., describes how autophagy promotes jasmonate-mediated defense against root-knot nematode (RKN) *Meloidogyne incognita*. The authors show that RKN induces autophagy in tomato and that autophagy deficient mutants are more susceptible to RKN infection. They further show that JAM1, a negative regulator of JA signaling, interacts with ATG8 in an AIM-dependent manner and is degraded via autophagy during infection. JAM1 negatively regulates JA signalling by disrupting the interaction between ERF1 and MED25. Finally, they show that ERF1 is positively regulating autophagy, by transcriptionally activating ATG8 gene expression. This constitutes a positive feedback loop, in which ERF1 activates autophagy during infection to remove negative regulator JAM1 in order to activate JA-mediated defense against RKN.

Overall, this study gives new insights in the role of autophagy during plant-microbe interactions and also describes a so far unknown crosstalk between autophagy and JA hormone pathways.

Comments/Specific points:

In the introduction, the authors describe the role of autophagy in plant-microbe interactions. However, they should update their literature for plant-bacteria interactions, how selective autophagy is able to counteract infection and also how microbes are manipulating autophagy by recent literature:

Üstün et al., 2018, <https://doi.org/10.1105/tpc.17.00815>

Lal et al., 2021 <https://pubmed.ncbi.nlm.nih.gov/32810441/>

Leong et al. 2022 <https://doi.org/10.15252/embj.2021110352>

Figure 1b: To assess autophagic activity/flux it is essential to include Concanamycin A treatment (or E64d) to block vacuolar degradation. Otherwise, induced autophagosomes might also indicate block of autophagy. Further increase of autophagosome-like structures should be expected if they block vacuolar degradation. The confocals are also not convincing, I hardly see any punctuate structures in their root images.

1d: The additional band might be also another isoform of ATG8. Thus, I propose to use the atg7 mutant used in the study together with the WT plant. If the additional band is indeed the lipidated form of ATG8, it should be absent in the atg7 mutant. This would strengthen the idea that autophagic degradation is activated during RKN infection.

1g: I am curious how the atg deficient tomato mutants look like. Do they produce fruits and seeds? Are they viable?

Figure 2: I wonder how a nuclear protein like JAM1 can interact with ATG8? Is it transported outside the nucleus upon autophagy induction? It would also be better to use AZD8055 treatment (TOR inhibitor/autophagy activator) instead of DTT, which might have additional effects.

Figure 3: As differences for treatments (e.g., E64d) are very small, I would suggest repeating this blot multiple times and quantify the band intensities. It might be also better to use ConA to block vacuolar degradation, as E64d is blocking cysteine proteases in general.

Figure 4d/e: I miss some in planta evidence here, e.g., BiFC and Co-IP in the presence or absence

of JAM1.

General comment: Is it possible that JAM1 is acting as a selective autophagy receptor for other proteins that it is interacting with? In Arabidopsis, MYC2 is interacting with MED25 and also other components. Have the authors thought about this possibility?

Reviewer #1:

The paper by Zou et al. describes a comprehensive analysis of the role of autophagy in JA-mediated plant defence against RKN in tomato. The authors have done a lot of work, using a various set of techniques and provide compelling evidence for the role of autophagy-mediated JAM degradation after RKN infection, mediated by interaction of JAMs with ATG8. They also nicely show how JAMs impair the ERF1-based activation of the JA signaling pathway that leads to plant defence responses against RKN. The work is novel and timely, and brings very interesting new insights for the field.

There are a few experimental elements and explanations missing. Below are my comments for improvement of the manuscript.

Response: Many thanks for your positive comments and suggestions. We have supplied some data which were not present in original version and have completed other experiments in response to your comments. Our response to your comments has been listed below point-by-point.

1/ The authors show a role for the JA-ERF1 branch in plant defence against RKN in tomato. However, this branch is typically associated with resistance to necrotrophic pathogens, while RKN are extremely biotrophic. How to reconcile this? Can this be discussed?

Response: This is a good question worth discussing. Traditionally, salicylic acid (SA) is a key component in plant defence against biotrophic pathogens, whereas the jasmonic acid (JA) and ethylene pathway acts mainly against necrotrophic pathogens and herbivores. Previous studies demonstrated that JA is a positive regulator in RKN and other nematode resistance, whilst SA had little or negative effects on the RKN resistance (Cooper, et al., 2005; Kammerhofer et al., 2015; Nahar et al., 2011). In agreement with these results, our previous results also found that tomato RKN resistance did not significantly differ between WT and SA deficient plants (*NahG*), but was decreased in JA deficient mutants (*spr2*) (Song et al., 2018). Thus, JA pathway is the main pathway to induce plant resistance against RKN. Based on the previous studies, we focus on the relationship between autophagy and JA pathway. Moreover, we also found JA-MYC2 branch negatively regulates tomato RKN resistance in our previous study (Xu et al., 2019). Thus, we have tested the function of ERF1 branch on tomato RKN resistance and have found that ERF1 branch positively regulates tomato RKN resistance.

2/ Line 120 is rather confusing. The authors state here that JA signaling is required for tomato susceptibility to RKNs. In the rest of their manuscript, however, they always link JA with

resistance, not susceptibility. See for example a few lines further (line 144) where they state that JA has a 'crucial position in RKN resistance'. This should be explained better in the text.

Response: We are sorry for this mistake. JA signaling pathway is required for tomato resistance to RKNs. We have corrected this mistake.

3/ It is not well-explained why the authors monitor expression of Pdf1.2a and Pdf1.2b in tomato after RKN infection. Why these genes? What does induction of these genes indicate? Are they used here as JA-marker genes? What is their role in tomato resistance against RKN?

Response: PDF1.2s are molecular markers of the JA-mediated defense response signaling pathways. The expression of *PDF1.2a* and *PDF1.2b* after RKN and other pathogen infection indicated plants' general resistance levels (Kammerhofer et al., 2015; Xu et al., 2019). In present study, we found that the expression of *PDF1.2a* and *PDF1.2b* was consistent with RKN resistance of different tomato genotypes and can reflect tomato RKN resistance. In manuscript, we have introduced that *PDF1.2a* and *PDF1.2b* are JA marker and responsive genes (Line 185-186).

4/ Susceptibility to RKN should be monitored in an ATG8-mutant. Now, four mutants in other ATG-genes are included. How were these 4 selected?

Response: ATG8 protein is a key player of the autophagy core machinery. The ATG8 protein is encoded by nine genes in Arabidopsis and seven genes in tomato. The function of these ATG8s are highly redundant. Thus, no phenotype had been reported so far for the different *ATG8* single mutants isolated in Arabidopsis, tomato and other plants. ATG4, ATG6, ATG7 and ATG10 which we selected to research in this study, all play important roles in the main proposed events in the autophagy pathway and are all present as single genes in tomato. ATG4 is a cysteine protease and functions as an essential factor in the ATG8 conjugation system. The C-terminal extension of newly synthesized ATG8 has to be cleaved by ATG4 protease to expose a highly conserved Gly, which is necessary for conjugation to PE. Additionally, ATG4 functions as a deconjugating enzyme that cleaves the amide bond between ATG8 and PE allowing the recycling of free ATG8 (Yu et al., 2012). ATG6 is a subunit of the class III phosphatidylinositol-3-kinase (PI3K) complex which remodels autophagic membranes. ATG7 is an ubiquitin-activating E1-like enzyme and is crucial for both the autophagy conjugation complex and the autophagosome formation. ATG7 can activate ATG8 and ATG12 to form ATG8-PE and ATG12-ATG5 complex. ATG10, an E2-like enzyme, is required for ATG12-ATG5 complex formation (Bassham et al., 2006). In plants, *ATG7* mutants

are commonly used in studies of autophagy deficiency (Thompson et al., 2005; Shin et al., 2014). Therefore, when we found that four different *ATG* mutants were sensitive for RKN infection, we selected *ATG7* to study the function of autophagy in tomato RKN resistance.

5/ the authors studied the interacting proteins of ATG8a using Y2H. This was however only done for interaction with plant proteins. Why not for nematode proteins? It would be extremely interesting to find out if RKN secretes effectors that potentially interact with ATG8s.

Response: This is a very good question and thanks for your suggestion. In this study, we focus on the function of autophagy on JA-mediated plant defence against RKN. Actually, we have constructed the yeast library of RKN effectors. Using ATG8a and selective autophagy receptor NBR1a/b as baits, we have found several RKNs secrete effectors to interact with ATG8a and NBR1a/b. We hypothesize that plant autophagy can degrade effectors secreted by RKNs to interfere with RKN infection. We hope we can complete the experiments and publish these stories in the near future.

6/ The use of DTT, as inducer of autophagy, should be used as positive control for the experiments shown in Fig. 1b and Fig. 1d.

Response: Based on your comments and Reviewer #2's comments, we have assessed autophagic activity after RKN infection by using Concanamycin A (ConA) treatment which block vacuolar degradation and further increase of autophagosome-like structures (Fig. 1b and 1d). ConA treatment blocked autophagic body degradation, which not only enhanced the microscopic detection of autophagosomes but also stabilized their content. We have also detected ATG8 and ATG8-PE in *atg7* mutants after RKN infection as a negative control (Fig. 1d). No ATG8-PE bands were detected regardless of whether the *atg7* mutants were infected with RKN. A weak ATG8-PE band was detected in WT roots under normal conditions. However, the signal increased after RKN infection. The results further demonstrate that infection of RKNs induces autophagy in tomato roots.

7/ Lines 225-238: why were these experiments done on the *ATG7* mutant? Why not on other *ATG* mutants? Why was this mutant selected?

Response: In plants, *atg7* mutants are commonly used in studies of autophagy deficiency (Thompson et al., 2005; Shin et al., 2014). Therefore, when we found that four different *atg* mutants were sensitive for RKN infection, we selected *atg7* mutants to study the function of autophagy in tomato RKN resistance.

8/ To demonstrate that JAM1 is regulated by the autophagic pathway, the authors do assays with chemical inhibitors. An additional experiment should also be done in nematode-infected plants, using the same chemicals.

Response: We supplied western blotting experiments to detect JAM1 accumulation after RKN infection using the same chemicals. Based on Reviewer #2's suggestion, we used ConA and AZD8055 instead of E64d and DTT, respectively (Fig. 3c & 3d). We repeated western blotting experiments three times and quantified the band intensities.

9/ Lines 273-276: the information provided here is based on Arabidopsis work. Is a one-to-one translation to tomato valid? Do the 2 JA-branches (MYC2 and ERF1) function similar in tomato as in Arabidopsis? Do the 2 branches differ in their role towards RKN? Apparently they do, but this should be better explained in the manuscript.

Response: MYCs and ERFs are two antagonistic branches in JA-mediated resistance towards different biotic stresses in plants. In general, the ERF branch is associated with enhanced resistance to necrotrophic pathogens, whereas the MYC branch is associated with wound response and defense against insect herbivores (Hickman et al., 2017). They may have different functions in response to different pathogens between Arabidopsis and tomato (Du et al., 2017). Our previous results support that MYC2 negatively regulates RKN resistance in tomato (Xu et al., 2019). In this study, we demonstrate that ERF1 is the major regulator in JA-mediated tomato RKN resistance. Thus, MYC2 and ERF1 are two antagonistic branches in JA-mediated RKN resistance in tomato. We further discuss their relationship in the Discussion (Line 491-495)

10/ How similar are the JAMs to MYC2?

Response: JAMs and MYC2 all belong to the group III bHLH protein family (Sasaki-Sekimoto et al., 2013). However, JAMs negatively regulate JA-mediated defense responses via impairing the formation of the MYC2-MED25 transcriptional activation complex and competing with the DNA binding capacity of MYC2 (Nakata et al., 2013; Sasaki-Sekimoto et al., 2013; Liu et al., 2019).

11/ The susceptibility of the *spr2* mutant to RKN should be presented in the manuscript.

Response: There are many studies have demonstrated that tomato JA deficient mutant *spr2* is susceptible for RKN infection (Fan et al., 2015; Song et al., 2018; Wang et al., 2019). In

present study, using *spr2* mutant, we mainly want to determine whether autophagy regulates JA biosynthesis and whether JA deficiency affects autophagy formation after RKN infection. Our results confirm that autophagy does not affect JA biosynthesis, but positively regulate *ERF1* expression and protein accumulation after RKN infection (Fig. 5). On the other side, JA deficiency blocks RKN-induced *ATGs* expression and autophagy formation (Fig. 6 & Supplementary Fig. 11). Based on the above reasons, we did not repeatedly detect the susceptibility of the *spr2* mutant to RKN.

12/ Line 390: this sentence seems to suggest that *ERF1* only has 3 target genes. This is however not known. The authors should monitor the expression of *ATG8a*, *ATG8b* and *ATG13* and other potential target genes (such as *pdf1.2a* & *b*) in the *erf1*-mutant using qRT-PCR. Alternatively, and RNA-seq study on this mutant could provide insights about potential other target genes of *ERF1*.

Response: In this manuscript, we found that *ERF1* can bind to the promoters of *PDF1.2a*, *PDF1.2b*, *ATG8a*, *ATG8b*, *ATG13*, and *ERF1* itself. Actually, we detected the expression of these genes and found that RKN-induced their transcripts were all compromised in *erf1* mutants (Fig. 4c for *PDF1.2a* and *PDF1.2b*; Supplementary Fig. 12 for *ATGs*). Of course, more potential target genes of *ERF1* can be found in future studies by using RNA-Seq, ChIP-Seq and other experiments.

13/ What is known about the function of the various homologues of *ATG8*?

Response: *ATG8* protein is a key player of the autophagy core machinery. Unlike a single *ATG8* in yeast, multiple *ATG8* homologues have been identified in the plants. There are nine *ATG8* homologues in *Arabidopsis* and seven *ATG8* homologues in tomato. These *ATG* proteins have been grouped into 3 groups based on amino acid sequence alignment (Kellner et al., 2017; Wang et al., 2020). In tomato, *ATG8a*, *ATG8c*, *ATG8d* and *ATG8f* belong to the first group, *ATG8b*, *ATG8e* and *ATG8g* belong to the second group, *ATG8h* and *ATG8i* belong to the third group, according to amino acid sequence alignment (Kellner et al., 2017). Different *ATG8s* have distinct spatial and temporal expression patterns in different tissues and are regulated by distinct factors (Wang et al., 2020). Meanwhile, plant *ATG8* expansion have been driven by functional diversification. For example, in present study, *JAMs* didn't interact with all *ATG8s* in yeast. They interact with the first group *ATG8a*, *ATG8d* and *ATG8f*, and the third group *ATG8h* by AIM motif (Fig. 2 & Supplementary Fig. 2), suggesting that the interaction between *JAMs* and *ATG8s* is not only related to the N-terminal AIM domain of

JAMs, but also is related to ATG8 structures. We have supplied the description and analysis of the ATG8s and JAMs interaction in the Results (Line 203-209, Line 232-234).

14/ Line 452: the 'inconsistency' reported here is not so surprising. The proteins are degraded by autophagy, not the mRNA.

Response: We have revised this sentence as your suggestion (Line 485).

15/ Discussion: Can the authors comment on the potential of interference by the nematode in autophagy? For example, by secreting effectors that interact with ATGs.

Response: We have discussed the potential of interference by the nematode in autophagy as followed: In addition, the discovery of the new functions of autophagy on plant-nematode interaction provide another interesting direction for future exploration. On the one hand, plant autophagy may reduce nematode infection by degrading nematode secretion effector proteins; on the other hand, nematode secretion effector proteins may trigger plant immunity to initiate autophagy, while also have the possibility to interfere with autophagy activity (Line 535-541).

16/ The temperature at which the infections were done (20-23°C) are very low for tomato and RKN. Is this correct?

Response: The perfect temperature for growing tomato plants is 20-25°C. The perfect temperature for RKN is 20-30°C. The perfect temperature for virus-induced gene silencing in tomato plants is 18-23°C. In summary, the growing temperature (20-23°C) is the most suitable temperature for this study.

17/ The hormone measurements were done on only 3 biological replicates. The SD of these results, shown in fig. 5a, cannot be correct. Hormone levels typically vary a lot between plants, hence the error bars are expected to be larger.

Response: We have built a well-established method for phytohormone measurements in our laboratory. We published more than 20 articles using this method for jasmonates and other phytohormones in *PNAS*, *Current Biol*, *Autophagy* and others. Similar results about JA contents in tomato roots with or without RKN infection were published in *Current Biology* (Wang et al., 2019) and *Plant Cell and Environment* (Song et al., 2018) by using 3-4 biological replicates. We are therefore confident in our data.

18/ The model shown in Fig. 6f seems incomplete. Negative interactions should be better indicated. Also the knowledge on MYC2 (based on a previous publication from this group) could be added to provide a better insight into the role of the two JA branches. What are the red and green arrows indicating?

Response: Thanks for your suggestion. We have improved the model. Taken together, our study demonstrated that autophagy positively regulated JA signaling and RKN resistance in tomato plants (Fig. 6f). In brief, autophagy promotes JA-mediated defense against RKNs by forming a positive feedback loop to degrade JAM1 and stimulate JA-ERF1 branch. JAM proteins interfere with the MED25-ERF1 interaction and DNA binding activity of ERF1 to interrupt JA-triggered defense responses. ERF1 functions as a transcriptional regulator for JA responsive genes. Furthermore, the expression of ATGs is regulated by JA signaling in an ERF1-dependent manner, and ERF1 expression is also self-regulated. In addition, the MYC2 branch may also interfere with the MED25-ERF1 complex to negatively regulate RKN resistance. The red blocking symbols indicate inhibition; green arrows indicate promotion.

Minor issues

- Line 51: write N-protein in full

Response: N protein is the name of tobacco mosaic virus resistance protein (Whitham, et al., 1994). It is entire.

- Line 200: ATG8a homologous proteins: reference needed

Response: We have supplied the literatures of ATG8 (Line 204).

- Line 374: ERF1 is critical for ATG (not JA)-mediated autophagy.

Response: We have revised this sentence. "Thus, ERF1 is critical for in RKN-induced autophagy formation and ATGs expression (Line 407-408)"

- Line 495: 'can precisely regulate JA signaling'. This statement is too vague

Response: We have revised this sentence. "In our study, we revealed the mechanism by which autophagy regulated tomato RKN resistance through promoting the JA signaling pathway, and elucidated that selective autophagy can degrade JA signaling negative regulators JAMs to active JA-ERF signaling branch, suggesting that autophagy may play complex but important roles in plant environmental responses (Line 529-533)"

Reviewer #2 (Remarks to the Author):

The present manuscript by Zou et al., describes how autophagy promotes jasmonate-mediated defense against root-knot nematode (RKN) *Meloidogyne incognita*. The authors show that RKN induces autophagy in tomato and that autophagy deficient mutants are more susceptible to RKN infection. They further show that JAM1, a negative regulator of JA signaling, interacts with ATG8 in an AIM-dependent manner and is degraded via autophagy during infection. JAM1 negatively regulates JA signalling by disrupting the interaction between ERF1 and MED25. Finally, they show that ERF1 is positively regulating autophagy, by transcriptionally activating ATG8 gene expression. This constitutes a positive feedback loop, in which ERF1 activates autophagy during infection to remove negative regulator JAM1 in order to activate JA-mediated defense against RKN.

Overall, this study gives new insights in the role of autophagy during plant-microbe interactions and also describes a so far unknown crosstalk between autophagy and JA hormone pathways.

Response: Many thanks for your positive comments and suggestions. We have supplied some data which were not present in original version and have completed other experiments in response to your comments. Our response to your comments has been listed below point-by-point.

Comments/Specific points:

In the introduction, the authors describe the role of autophagy in plant-microbe interactions. However, they should update their literature for plant-bacteria interactions, how selective autophagy is able to counteract infection and also how microbes are manipulating autophagy by recent literature:

Üstün et al., 2018, <https://doi.org/10.1105/tpc.17.00815>

Lal et al., 2021 <https://pubmed.ncbi.nlm.nih.gov/32810441/>

Leong et al. 2022 <https://doi.org/10.15252/emj.2021110352>

Response: In the introduction, we have updated describe the role of autophagy in plant-microbe interactions (Line 55-78).

Figure 1b: To assess autophagic activity/flux it is essential to include Concanamycin A treatment (or E64d) to block vacuolar degradation. Otherwise, induced autophagosomes might also indicate block of autophagy. Further increase of autophagosome-like structures should be expected if they block vacuolar degradation. The confocals are also not convincing,

I hardly see any punctate structures in their root images.

Response: In Fig. 1b, we supplied ConA treatment to assess autophagic activity after RKN infection. The results showed that autophagosome-like structures were further increased with ConA treatment after RKN infection.

1d: The additional band might be also another isoform of ATG8. Thus, I propose to use the ATG7 mutant used in the study together with the WT plant. If the additional band is indeed the lipidated form of ATG8, it should be absent in the ATG7 mutant. This would strengthen the idea that autophagic degradation is activated during RKN infection.

Response: In Fig. 1d, we have detected the ATG8 and ATG8-PE bands using WT, *atg7* seedlings after RKN infection. We also supplied ConA treatment at 48 hpi. The results showed that RKN-induced ATG8-PE bands were compromised in *atg7* mutants and ATG8 and ATG8-PE bands were more accumulated with ConA treatment in AC plants.

1g: I am curious how the ATG deficient tomato mutants look like. Do they produce fruits and seeds? Are they viable?

Response: In tomato, we observed that under normal growth conditions, the growth of these four *atg* mutants and WT plants was relatively consistent, and there was no significant difference in plant height, stem diameter, and photosynthesis. However, autophagy deficiency delayed fruit ripening (delayed 2~6 days) and reduced the number of seeds (reduced by 40~70%) in tomato fruit. These data have been prepared and will be submitted.

Figure 2: I wonder how a nuclear protein like JAM1 can interact with ATG8? Is it transported outside the nucleus upon autophagy induction? It would also be better to use AZD8055 treatment (TOR inhibitor/autophagy activator) instead of DTT, which might have additional effects.

Response: JAMs are transcription factors, however, these proteins are localized not only in the nucleus but also in the cytoplasm (Supplementary Fig. 3). We can detect BiFC signals from the interaction of JAMs-nYFP and cYFP-ATG8a and they are overlapped with mCherry-ATG8f signals (Fig. 2c & Supplementary Fig. 3). According to your suggestion, we used AZD8055 treatment instead of DTT and found that the numbers of punctate fluorescent signals of JAMs-ATG8a BiFC and mCherry-ATG8f were both sharply increased and were strongly overlapped (Fig. 2c & Supplementary Fig. 4).

Figure 3: As differences for treatments (e.g., E64d) are very small, I would suggest repeating this blot multiple times and quantify the band intensities. It might be also better to use ConA to block vacuolar degradation, as E64d is blocking cysteine proteases in general.

Response: According to your suggestion, we used ConA and AZD8055 treatments instead of E64d and DTT treatments, respectively. We also detected JAM1 accumulation after RKN infection according to Reviewer #1's suggestion (Fig. 3b & Source Data). We also repeated western blotting assays of JAM1 accumulation three times which showed similar results and quantify the band intensities (Fig. 3c, d & Source Data).

Figure 4d/e: I miss some in planta evidence here, e.g., BiFC and Co-IP in the presence or absence of JAM1.

Response: We supplied Co-IP assays to further confirm that JAM1 interfere with ERF1 and MED25 interaction (Fig. 4f).

General comment: Is it possible that JAM1 is acting as a selective autophagy receptor for other proteins that it is interacting with? In Arabidopsis, MYC2 is interacting with MED25 and also other components. Have the authors thought about this possibility?

Response: These are very good questions. JAM1 may be a potential selective autophagy receptor for their interacting proteins under other conditions. However, in this study, JAM1 is degraded by autophagy and *jams* mutants show increased formation of autophagosomes and higher resistance after nematode infection. Therefore, JAMs serve as the substrates of selective autophagy after nematode infection, rather than cargo receptors responsible for the degradation of other substrates in tomato after RKN infection.

MYC2 is interacting with MED25 in both Arabidopsis and tomato (Liu et al., 2019). Our previous study found that MYC2 negatively regulated RKN resistance in tomato (Xu et al., 2019). However, tomato MYC2 is not interacting with ATG8s by Y2H and BiFC assays (data not shown). Therefore, MYC2 may interfere with the ERF1-MED25 complex and ERF1-mediated RKN resistance. We have added MYC2 function in Discussion and model (Fig. 6f).

References

- 1) Cooper, W. R., Jia, L. & Goggin, L. Effects of jasmonate-induced defenses on root-knot nematode infection of resistant and susceptible tomato cultivars. *J. Chem Ecol* **31**, 1953-1967 (2005).
- 2) Kammerhofer, N. *et al.* Role of stress-related hormones in plant defence during early infection of the cyst nematode *Heterodera schachtii* in Arabidopsis. *New Phytol* **207**, 778-789 (2015).
- 3) Nahar, K. *et al.* The jasmonate pathway is a key player in systemically induced defense against root

- knot nematodes in rice. *Plant Physiol* **157**, 305-316 (2011) .
- 4) Song, L. X. *et al.* Brassinosteroids act as a positive regulator for resistance against root-knot nematode involving respiratory burst oxidase homolog-dependent activation of MAPKs in tomato. *Plant Cell Environ* **41**, 1113-1125 (2018).
 - 5) Kammerhofer, N. *et al.* Systemic above- and belowground cross talk: hormone-based responses triggered by *Heterodera schachtii* and shoot herbivores in *Arabidopsis thaliana*. *J Exp Bot* **66**, 7005-7017 (2015).
 - 6) Xu, X. C. *et al.* Strigolactones positively regulate defense against root-knot nematodes in tomato. *J Ex Bot* **70**, 1325-1337 (2019).
 - 7) Yu, Z. Q. *et al.* Dual roles of ATG8-PE deconjugation by ATG4 in autophagy. *Autophagy* **8** 883-892 (2012).
 - 8) Bassham, D. C. *et al.* Autophagy in development and stress responses of plants. *Autophagy* **2**, 2-11 (2006).
 - 9) Thompson, A. R., Doelling, J. H., Suttangkakul, A. & Vierstra, R. D. Autophagic nutrient recycling in *Arabidopsis* directed by the ATG8 and ATG12 conjugation pathways. *Plant Physiol* **138**, 2097-2110 (2005).
 - 10) Shin, K. D., Lee, H.N. & Chung, T. A revised assay for monitoring autophagic flux in *Arabidopsis thaliana* reveals involvement of Autophagy-related9 in autophagy. *Mol. Cells* **37**, 399-405 (2014).
 - 11) Hickman, R. *et al.* Architecture and dynamics of the jasmonic acid gene regulatory network. *Plant Cell* **29**, 2086-2105 (2017).
 - 12) Du, M. M. *et al.* MYC2 orchestrates a hierarchical transcriptional cascade that regulates jasmonate-mediated plant immunity in tomato. *Plant Cell* **29**, 1883-1906 (2017).
 - 13) Nakata, M., *et al.* A bHLH-type transcription factor, ABAINDUCTIBLE BHLH-TYPE TRANSCRIPTION FACTOR/JA-ASSOCIATED MYC2-LIKE1, acts as a repressor to negatively regulate jasmonate signaling in *Arabidopsis*. *Plant Cell* **25**, 1641-1656 (2013).
 - 14) Sasaki-Sekimoto, Y. *et al.* Basic helix-loop-helix transcription factors Jasmonate-Associated MYC2-LIKE1 (JAM1), JAM2, and JAM3 are negative regulators of jasmonate responses in *Arabidopsis*. *Plant Physiol* **163**, 291-304 (2013) .
 - 15) Liu, Y. Y. *et al.* MYC2 regulates the termination of jasmonate signaling via an autoregulatory negative feedback loop. *Plant Cell* **31**, 106-127 (2019).
 - 16) Fan, J. W. *et al.* Jasmonic acid mediates tomato's response to root knot nematodes. *J Plant Growth Regul* **34**, 196-205 (2015).
 - 17) Wang, G. T. *et al.* Systemic root-shoot signaling drives jasmonate-based root defense against nematodes. *Curr Biol* **29**, 3430-3438 (2019).
 - 18) Kellner, R. *et al.* ATG8 expansion: A driver of selective autophagy diversification? *Trends Plant Sci* **22**, 204-214 (2017).
 - 19) Wang, P. Nolan, T. M. Yin Y & Bassham, D. C. Identification of transcription factors that regulate ATG8 expression and autophagy in *Arabidopsis*. *Autophagy* **16**, 123-139 (2020).
 - 20) Haxim, Y. *et al.* Autophagy functions as an antiviral mechanism against geminiviruses in plants. *Elife* **6**

(2017).

- 21) Hafren, A. *et al.* Selective autophagy limits cauliflower mosaic virus infection by NBR1-mediated targeting of viral capsid protein and particles. *P Natl Acad Sci USA* **114**, E2026-E2035 (2017).
- 22) Hafren, A. *et al.* Turnip mosaic virus counteracts selective autophagy of the viral silencing suppressor HCpro. *Plant Physiol* **176**, 649-662 (2018).
- 23) Cao, B., Ge, L., Zhang, M., Li, F. & Zhou, X. Geminiviral C2 proteins inhibit active autophagy to facilitate virus infection by impairing the interaction of ATG7 and ATG8. *J Integr Plant Biol* doi:10.1111/jipb.13452 (2023).
- 24) Lal, N. K. *et al.* Phytopathogen effectors use multiple mechanisms to manipulate plant autophagy. *Cell Host Microbe* **28**, 558-571 (2020).
- 25) Dagdas, Y. F. *et al.* An effector of the Irish potato famine pathogen antagonizes a host autophagy cargo receptor. *Elife* **5**, 10856 (2016).
- 26) Whitham, S. *et al.* The product of the tobacco mosaic virus resistance gene *N*: similarity to toll and the interleukin-1 receptor. *Cell* **78**, 1101-1115 (1994).
- 27) Liu, Y. Y. *et al.* MYC2 regulates the termination of jasmonate signaling via an autoregulatory negative feedback loop. *Plant Cell* **31**, 106-127 (2019).

Reviewer #1 (Remarks to the Author):

The authors have addressed the majority of my comments. However, there are still 1 comment that needs to be resolved before a publication in a high-impact journal as Nature Communications is warranted. The data on nematode interaction partners of ATG8 and NBR1a/b should be included here to increase the impact of the reported findings.

Reviewer #2 (Remarks to the Author):

Overall i am satisfied with the changes. There are only tqo small issues:

-The actin blot in figure 3C does not correspond to the Jam1 blot (9 versus 8 samples). Please carefully check this data again. Maybe it would be generally good to provide the original blots as a raw data file to avoid problems like this.

-Additionally the new confocals in Fig 2c and Supp. 4a and b are of poor quality and need some improvement.

Otherwise congrats to the authors for this interesting story.

Reviewer #1:

The authors have addressed the majority of my comments. However, there are still 1 comment that needs to be resolved before a publication in a high-impact journal as Nature Communications is warranted. The data on nematode interaction partners of ATG8 and NBR1a/b should be included here to increase the impact of the reported findings.

Response: Many thanks for your comments and suggestions. In this study, we focus on the function of plant autophagy on JA signaling pathway and JA-mediated plant defence against RKN. The data on the interaction between nematode secretory proteins/effectors and ATG8/NBR1s and the degradation of nematode secretory proteins/effectors by plant autophagy are not related to the main story of this article, so we don't think these data should be included in this article. However, based on the importance of this research direction, we have conducted an in-depth discussion on the new functions of autophagy in plant-nematode interactions (Line 535-541).

Reviewer #2:

Overall i am satisfied with the changes. There are only tqo small issues:

- The actin blot in figure 3C does not correspond to the Jam1 blot (9 versus 8 samples). Please carefully check this data again. Maybe it would be generally good to provide the original blots as a raw data file to avoid problems like this.
- Additionally the new confocals in Fig 2c and Supp. 4a and b are of poor quality and need some improvement.

Otherwise congrats to the authors for this interesting story.

Response: Many thanks for your comments and suggestions. We are sorry for this mistake in Fig. 3c. We have corrected this mistake and provided the original blots in Source Data file (Fig. 3c). We have also improved the quality of confocals in Fig. 2c and Supplementary Fig. 4a.